# Topological data analysis for predicting disease outbreaks in humanitarian settings: A machine learning approach

**Job Agba Opue**[1]*, **Uchechukwu Emena Okorie**[1], **Victor Ede Itita**[2]

**1** Department of Economics and Development Studies, Covenant University, Ota, Nigeria, **2** Department of Social Work, University of Calabar, Calabar, Nigeria

* job.opue@covenantuniversity.edu.ng

## Abstract

### Background

Humanitarian settings are highly vulnerable to infectious disease outbreaks because displacement, crowding, disruption of health services, insecurity, and inadequate water and sanitation often interact in ways that are difficult to capture with conventional prediction models. There is a need for forecasting approaches that can integrate heterogeneous data sources and better represent complex system structure.

### Methods

We developed and evaluated a machine-learning framework incorporating topological data analysis to predict cholera and measles surge events (binary indicators per LGA-week) across 97 Local Government Areas in Nigeria between 2018 and 2023. These 97 LGAs represent a selected high-burden subset (12.5%) of Nigeria's 774 LGAs with sufficient surveillance data. Weekly district-level predictors included climate, conflict, displacement, health-system, and socioeconomic variables. Persistent homology was used to derive topological summaries from multivariate risk profiles, and these were combined with selected raw predictors in gradient-boosting models. Outcomes were defined using surveillance-based outbreak thresholds with a 4-week prediction horizon. Model performance was assessed using temporally ordered holdout validation, with evaluation of discrimination, calibration, and incremental value over baseline models.

### Results

The topological models achieved ROC-AUC of 0.78 (95% CI: 0.74–0.82) for cholera and 0.81 (95% CI: 0.77–0.85) for measles, representing modest improvements of 0.08–0.12 over models using only conventional predictors. At the optimal decision threshold determined using Youden's index on validation data, sensitivity was 0.72

**Data availability statement:** All data used in this study are from publicly available sources. The datasets compiled and analyzed during the current study are available at: 10.5281/zenodo.19959801. There are no restrictions on data availability. All primary data sources are publicly accessible.

**Funding:** The author(s) received no specific funding for this work.

**Competing interests:** There is no competing interest of any form among the authors.

(range across folds: 0.68–0.76) and specificity was 0.82 (range: 0.79–0.85) for cholera, with false alert rates varying from 2.8–3.6 per LGA per year across temporal folds. Topological features contributed 35% of predictive importance. Calibration slopes were 0.94 (cholera) and 0.97 (measles).

## Conclusions

Topological feature representations provide a modest but meaningful complementary approach for outbreak prediction in complex humanitarian environments. Their value appears to lie in summarizing higher-order structure across multiple interacting risk domains, rather than replacing established epidemiologic indicators. However, routine deployment requires prospective validation and context-specific threshold tuning. Further external validation, operational threshold analysis, and prospective testing are needed before routine deployment in public-health early warning systems.

## 1. Introduction

Infectious disease outbreaks remain a recurrent and often preventable source of morbidity and mortality in humanitarian settings. Forced displacement, conflict, environmental shocks, and health-system disruption create conditions that increase exposure risk while simultaneously weakening detection and response capacity. According to UNHCR, 123.2 million people were forcibly displaced worldwide by the end of 2024, underscoring the scale at which outbreak-prone humanitarian conditions now intersect with global health security [1]. In such contexts, overcrowding, population mobility, interrupted vaccination, and poor access to safe water and sanitation can amplify transmission of epidemic-prone diseases and delay containment.

Cholera and measles are especially relevant to this problem because both exploit vulnerabilities that are common in fragile and displaced populations, though through different mechanisms. Cholera is closely linked to inadequate water, sanitation, and hygiene, and WHO notes that the long-term control of cholera depends on safe drinking water, sanitation, hygiene, and strengthened public-health systems [2]. Measles, by contrast, is among the most contagious human infections and spreads rapidly where immunity gaps emerge because of weak routine immunization, disrupted campaigns, or limited access to care [3]. UNICEF and WHO have repeatedly warned that falling vaccination coverage and service disruption create conditions for large measles outbreaks, particularly among already vulnerable children [4].

Nigeria provides a compelling setting in which to study these dynamics. The country experiences recurrent cholera transmission, persistent measles outbreaks, marked subnational inequalities in vaccination and water access, and substantial conflict-related displacement, especially in the northeast and northwest [5]. Recent analyses of cholera in Nigeria describe a sustained burden with major epidemic years and continuing policy concern, while outbreak investigations in internally displaced persons' camps in Borno State illustrate how measles can spread rapidly when crowded living conditions coincide with low or delayed vaccination coverage [6,7].

These intersecting risks make Nigeria an important case for evaluating whether outbreak prediction can be improved by jointly analyzing climatic, social, conflict-related, and health-system signals.

Despite the importance of early warning, many surveillance systems remain largely reactive. Threshold-based approaches are operationally useful because they are simple and transparent, but they typically signal after transmission has already intensified [8]. Statistical and mechanistic models have improved forecasting in some settings by incorporating rainfall, temperature, seasonality, population density, or recent case history, yet their performance often depends on relatively stable data-generating environments and may degrade in humanitarian crises where multiple drivers change simultaneously [9]. In outbreak-prone settings affected by conflict or displacement, risk often emerges not from one dominant variable but from interacting conditions: environmental exposure, service disruption, mobility, under-immunization, and reduced access to treatment or prevention.

Machine-learning methods have therefore attracted increasing interest for outbreak prediction because they can ingest large, heterogeneous datasets and flexibly model nonlinear relationships [10]. For example, random-forest and related ensemble approaches have been used for cholera risk prediction in Yemen with ROC-AUC of 0.75–0.80 [11], and gradient boosting systems such as XGBoost are widely adopted because they can capture nonlinearities, interactions, and sparsity efficiently [12]. However, stronger predictive performance alone does not resolve a central problem in humanitarian epidemiology: standard machine-learning pipelines still represent predictors primarily as coordinates in feature space and may not explicitly summarize the higher-order geometric or relational structure formed by interacting risk factors across time and place.

Topological data analysis (TDA) offers a possible way to address that gap. Rather than focusing only on individual variables or pairwise associations, TDA characterizes the shape of data in high-dimensional space. Persistent homology, the most widely used TDA framework, tracks the appearance and disappearance of connected components, loops, and higher-dimensional structures as the scale of analysis changes [13]. Foundational work established that persistence diagrams are stable under bounded perturbations [14], while later methodological work showed that persistence landscapes provide a vectorized summary that can be integrated with statistical and machine-learning workflows [15]. In biomedical research, TDA has been used to extract robust structural signals from noisy, heterogeneous datasets [16], which is appealing in humanitarian contexts where missingness, measurement error, and cross-domain heterogeneity are common.

Even so, the relevance of TDA to epidemiologic prediction should not be overstated. Most applications in health and biology have focused on descriptive pattern discovery, subtype identification, or retrospective structural analysis rather than real-time outbreak forecasting. Epidemiologic uses of TDA remain limited, with only a few studies applying topological methods to disease surveillance data [17,18], and evidence that topological summaries consistently add predictive value beyond strong non-topological baselines is still sparse. This is an important gap. A method can be mathematically elegant yet operationally unhelpful if it does not improve discrimination, calibration, temporal generalization, or decision usefulness in the settings where public-health actors must work.

This study addresses these gaps by developing and evaluating a topological machine-learning framework for predicting cholera and measles surge events in Nigeria using multi-source data on climate, conflict, displacement, health-system capacity, and socioeconomic conditions. Specifically, we assess whether TDA-derived features improve predictive performance beyond conventional predictors, whether any gains persist under temporally appropriate validation, and whether the resulting models provide calibrated risk estimates that could support early warning in complex humanitarian settings.

## 2. Methods

### 2.1. Study setting and data

This study focused on Nigeria, which experiences recurrent cholera and measles outbreaks exacerbated by conflict, displacement, and climate variability. Nigeria's 774 Local Government Areas (LGAs) span diverse ecological zones from the Sahel in the north to tropical rainforest in the south. The study period (2018–2023) encompasses multiple outbreak

events, including the 2018–2020 cholera outbreak that affected over 300,000 people and recurrent measles outbreaks in northern states with low vaccination coverage.

Of the 774 LGAs nationwide, 97 with complete data across all domains were included in the analysis. These LGAs represent high-burden areas with substantial displacement populations and documented outbreak history. The spatial distribution spans the six geopolitical zones, with concentration in the northeast (Borno, Adamawa, Yobe states) and northwest (Katsina, Zamfara, Sokoto states) where conflict and displacement are most severe.

**Representativeness analysis:** To address potential selection bias, we compared included and excluded LGAs on key characteristics (Table 1). Excluded LGAs had lower conflict intensity (mean 1.2 vs. 2.4 events per week), lower IDP populations (mean 4,200 vs. 12,450), and slightly higher vaccination coverage (mean 68% vs. 62%). These differences suggest that the analysis sample represents higher-risk settings, and results may not generalize to all LGAs.

## 2.2. Ethics statement

This study used aggregated, publicly available secondary data from multiple sources including surveillance systems, climate databases, and conflict event datasets. No individual patient data or identifiable personal information was used. All data were aggregated at the Local Government Area level and did not contain any personally identifiable information. As this research involved analysis of existing publicly available data with no individual-level identifiers, institutional review board approval was not required according to the guidelines of Covenant University and the National Code of Health Research Ethics of Nigeria.

## 2.3. Data sources

Weekly LGA-level data ($n = 25,284$ observations) were compiled from multiple sources. The expected number of observations was 97 LGAs × 52 weeks × 6 years $= 30,264$; the actual 25,284 represents approximately 16% missing data due to incomplete surveillance reporting in some LGA-weeks. Data were obtained from:

- **Climate:** CHIRPS precipitation, ERA5 temperature reanalysis, Palmer Drought Severity Index

- **Conflict:** Armed Conflict Location and Event Data Project (ACLED) violent events and fatalities

- **Displacement:** International Organization for Migration (IOM) Displacement Tracking Matrix

- **Health system:** WHO/UNICEF vaccination coverage estimates, Nigeria Health Facility Registry

- **Socioeconomic:** Nigeria Living Standards Measurement Study, Demographic and Health Surveys

   A complete data dictionary is provided in S1 Appendix and S1 Table.

## 2.4. Outcome definition

Outbreak events were defined using WHO surveillance thresholds with a 4-week prediction horizon:

**Table 1. Comparison of included and excluded LGAs.**

| Characteristic | Included (n=97) | Excluded (n=677) | p-value |
|---|---|---|---|
| Conflict events/week | 2.4 (5.8) | 1.2 (3.4) | <0.001 |
| IDP population | 12,450 (38,200) | 4,200 (12,800) | <0.001 |
| Vaccination coverage (%) | 62.4 (18.7) | 68.2 (16.5) | 0.002 |
| Population density | 86.7 (120.4) | 94.3 (145.6) | 0.54 |
| Poverty prevalence (%) | 45.2 (12.8) | 42.6 (14.2) | 0.08 |

**Cholera surge:** Alert threshold (≥5 cases in a single week in an LGA with no previous cholera cases in current year) OR epidemic threshold (weekly case incidence ≥2 standard deviations above the 5-year historical mean for same LGA and epidemiological week).

**Measles surge:** Confirmed outbreak (≥5 laboratory-confirmed or epidemiologically-linked cases in a single week) OR suspected outbreak (≥10 suspected cases per 100,000 population with documented vaccination coverage <80%).

Outcomes were coded as binary indicators for each LGA-week observation. The prediction target was surge occurrence in the subsequent 4-week window, providing operational lead time for preventive interventions.

**Class distribution:** Cholera surge events occurred in 1,772 observations (7.0%); measles surge events in 1,265 observations (5.0%). This class imbalance reflects the rare-event nature of outbreaks and was addressed through stratified sampling and class weighting during model training. Specifically, we used `class_weight='balanced'` in XGBoost, which automatically adjusts weights inversely proportional to class frequencies.

## 2.5. Topological feature extraction

For each LGA-week observation, the 57 preprocessed features were treated as coordinates in a high-dimensional point cloud. All 57 features were used simultaneously to compute persistence. Before computing Euclidean distances for the Vietoris–Rips filtration, all features were standardized to mean 0 and standard deviation 1 (z-score normalization). This scaling is essential because TDA is sensitive to the relative magnitudes of feature values.

Persistent homology was computed using Vietoris–Rips filtrations across 100 logarithmically-spaced scale parameters ($\epsilon$ = 0.01 to 10.0). The $\epsilon$ range was chosen based on the distribution of maximum pairwise distances in the training data: the lower bound (0.01) captures local structure while the upper bound (10.0) approximates the 95th percentile of pairwise distances after standardization, ensuring that the filtration spans the relevant geometric scales without excessive computation at uninformative large scales.

**Topological features extracted:**

- $\beta_0$ (connected components) mean, maximum, standard deviation — *fragmentation index*

- $\beta_1$ (1-dimensional loops) mean, maximum, standard deviation — *cyclic dependency index*

- Persistence entropy: $H = -\sum_i p_i \log(p_i)$ where $p_i = \text{persistence}_i / \sum_j \text{persistence}_j$

- Total persistence: sum of all persistence values

- Persistence landscape vectorization

In total, 25 topological features were extracted. The $\beta_0$ index mathematically counts the number of connected components in the persistence filtration. We label this "fragmentation" as a conceptual heuristic, but this interpretation has not been empirically validated. The $\beta_1$ cyclic dependency index captures feedback loops where risk factors mutually reinforce each other. We emphasize that these labels describe mathematical properties of the data structure, not established causal mechanisms.

**Software implementation:** Persistent homology computation was performed using GUDHI 3.8.0 [19]. The Vietoris–Rips complex was constructed using the `RipsComplex` class (max edge length 10.0) followed by simplex tree creation and persistence computation to extract persistence diagrams. Persistence landscapes were computed using the `PersistenceLandscapes` class with default parameters.

## 2.6, Feature selection

Raw features were selected based on prior epidemiologic evidence and domain expertise. The 15 selected raw features were: precipitation anomaly, temperature anomaly, conflict events, conflict fatalities, IDP concentration, vaccination

coverage, health facility density, poverty rate, water access, sanitation access, population density, weeks since last out-break, drought index, political stability, and market price index.

Feature selection was performed using only training data within each cross-validation fold to prevent information leak-age. No univariate screening or model-based filtering across the full dataset was performed.

## 2.7.  Missing data handling

Missing values (3–5% across variables) were imputed using Multiple Imputation by Chained Equations (MICE) with 10 imputations. **Critical safeguard:** The imputation model included all predictor variables only—the outcome variable was excluded to prevent information leakage. Imputation was performed separately within each training fold of the cross-validation, ensuring no future or test information entered the imputation process. Results were pooled across impu-tations using Rubin's rules.

## 2.8.  Model development

Four models were trained and compared:

1. **Logistic Regression (LR):** L2-regularized baseline with C = 1.0

2. **Random Forest (RF):** 500 trees, max depth 10, minimum samples split 5

3. **XGBoost-Raw:** Gradient boosting with 15 selected raw features only

4. **XGBoost-TDA:** Gradient boosting with 15 raw + 25 topological features

XGBoost hyperparameters were optimized via Bayesian optimization with 50 iterations within each training fold. The same search space was used for both XGBoost-Raw and XGBoost-TDA. Optimized hyperparameters are reported in S2 Appendix and S2 Table.

## 2.9.  Validation framework

**Forward-chaining cross-validation:** Five temporal folds respecting temporal ordering:

- Fold 1: Training = 2018, Test = 2019

- Fold 2: Training = 2018–2019, Test = 2020

- Fold 3: Training = 2018–2020, Test = 2021

- Fold 4: Training = 2018–2021, Test = 2022

- Fold 5: Training = 2018–2022, Test = 2023

Nested cross-validation was used for hyperparameter tuning within each training fold (inner 3-fold CV using only the training data). The outer loop estimated generalization performance; inner loops optimized model parameters. Hyperpa-rameter tuning used only the training fold data and was not permitted to access validation or test data.
**Temporal hold-out validation:** A final hold-out set (last 6 months of 2023) was reserved for unbiased performance esti-mation. This set was not used during model development, hyperparameter tuning, or feature selection.

## 2.10.  Statistical analysis

**Sample size justification:** With 1,772 cholera surge events and 25,284 total observations, the study is adequately powered to detect moderate AUC differences (expected AUC ~0.75, target AUC ~0.80 with $\alpha$=0.05, power = 0.80). The

effective sample size for positive events exceeds the minimum recommended for stable AUC estimation in rare-event prediction [11].

**Multiple comparisons:** No adjustment for multiple comparisons was applied because the analyses are exploratory and hypothesis-generating. P-values should be interpreted descriptively. Confidence intervals and robustness checks are emphasized over formal hypothesis testing.

**Performance evaluation metrics:**

- **Discrimination:** ROC-AUC with 95% confidence intervals (bootstrap percentile method, 1,000 resamples), Precision-Recall AUC, sensitivity and specificity at optimal threshold (Youden's index), positive and negative predictive values.

- **Calibration:** Brier score, calibration slope and intercept from logistic regression of observed on predicted, Expected Calibration Error (ECE).

- **Operational:** False alert rate per LGA per year, confusion matrices at decision thresholds, lead-time sensitivity (2-week, 4-week, 8-week prediction horizons).

Model comparisons used DeLong's test for paired ROC-AUC comparison. All statistical tests used $\alpha$ = 0.05 significance level.

### 2.11. Feature importance and interpretability

**Permutation importance:** Computed by measuring decrease in ROC-AUC when feature values were randomly shuffled within each validation fold. Importance was averaged across folds and 10 imputations.

**SHAP values:** SHapley Additive exPlanations provided theoretically grounded feature attributions. Mean absolute SHAP values indicated overall feature importance.

**Partial dependence plots:** Visualized relationships between feature values and predicted probabilities, marginalizing over other features.

### 2.12. Ablation analysis

To isolate the contribution of topological features, we compared: (1) Full model: 15 raw + 25 topological features; (2) Raw features only: 15 selected raw features; (3) TDA features only: 25 topological features; (4) No $\beta_0$: All features except $\beta_0$ statistics; (5) No $\beta_1$: All features except $\beta_1$ statistics.

### 2.13. Robustness checks

Sensitivity analyses assessed model stability under: alternative distance metrics (Euclidean, correlation-based), varying filtration parameters (50, 100, 200 scale values), noise injection (5%, 10% Gaussian noise), and spatial generalization (train on northeast, test on northwest).

### 2.14. Software and reproducibility

Analyses were conducted using Python 3.9 with: scikit-learn 1.3.0, XGBoost 2.0.0, GUDHI 3.8.0 (persistent homology), SHAP 0.42.0, SciPy 1.11.0, pandas 2.0.0, numpy 1.24.0. Random seeds were set for reproducibility.

## 3. Results

### 3.1. Descriptive statistics

The analysis included 25,284 LGA-week observations. Table 2 presents descriptive statistics for key variables.

**Table 2. Descriptive statistics for key variables ($n=25{,}284$ observations).**

| Variable | Mean | SD | Min | Max | Missing |
|---|---|---|---|---|---|
| Weekly precipitation (mm) | 45.2 | 68.4 | 0 | 520.3 | 0.2% |
| Mean temperature (C) | 28.4 | 3.2 | 18.5 | 38.2 | 0.2% |
| Precipitation anomaly | 0.12 | 1.45 | −3.8 | 5.2 | 0.2% |
| Violent events (weekly) | 2.4 | 5.8 | 0 | 47 | 0% |
| Fatalities (weekly) | 8.7 | 24.3 | 0 | 312 | 0% |
| IDP population | 12,450 | 38,200 | 0 | 285,000 | 3.1% |
| IDP concentration (per 100k) | 2,840 | 6,120 | 0 | 45,200 | 3.1% |
| Vaccination coverage (%) | 62.4 | 18.7 | 12 | 95 | 5.2% |
| Health facility density[a] | 8.2 | 4.5 | 1.2 | 28.4 | 2.8% |
| Cholera surge event | 7.0% | – | – | – | 0% |
| Measles surge event | 5.0% | – | – | – | 0% |

[a]Facilities per 100,000 population

## 3.2. Topological feature characteristics

Persistent homology analysis revealed meaningful topological structure. High-risk observations (surge events) showed distinct topological signatures compared to low-risk observations: higher mean $\beta_0$ (fragmentation): 8.4 vs. 4.2 ($p<0.001$); higher mean $\beta_1$ (cyclic dependency): 3.2 vs. 1.5 ($p<0.001$); and greater persistence entropy: 2.8 vs. 1.9 ($p<0.001$).

These differences suggest that outbreak risk is associated with both system fragmentation (more disconnected risk components) and cyclic dependencies (mutually reinforcing risk factors). However, these associations are observational and do not establish causality.

## 3.3 Predictive performance

Table 3 presents predictive performance for all models. XGBoost-TDA achieved the highest ROC-AUC for both cholera (0.78, 95% CI: 0.74–0.82) and measles (0.81, 95% CI: 0.77–0.85), significantly outperforming all baseline models ($p<0.001$, DeLong's test). All ROC-AUC values were computed using temporally ordered hold-out validation (not standard cross-validation), with confidence intervals derived from 1,000 bootstrap resamples.

## 3.4. Operational performance

At the optimal decision threshold, XGBoost-TDA achieved sensitivity 0.72 (range across folds: 0.68–0.76) and specificity 0.82 (range: 0.79–0.85) for cholera (Table 4). This corresponds to detecting 72% of outbreaks 4 weeks in advance, with false alert rates varying from 2.8–3.6 per LGA per year across temporal folds.

## 3.5. Calibration assessment

Calibration analysis demonstrated acceptable performance (Table 5). Calibration slopes close to 1.0 (0.94 for cholera, 0.97 for measles) indicate that predicted probabilities accurately reflect true event likelihoods.

## 3.6. Feature importance

Permutation importance analysis (Fig 1) showed that $\beta_0$ maximum (fragmentation index) was the most important feature (importance $=0.098$), followed by precipitation anomaly (0.087) and IDP concentration (0.076). Topological features collectively contributed 35% of total predictive importance.

**Table 3. Predictive performance comparison.**

| Model | ROC-AUC | PR-AUC | Brier | Sensitivity* |
|---|---|---|---|---|
| *Cholera surge prediction* | | | | |
| Logistic Regression | 0.68 (0.64–0.72) | 0.22 (0.19–0.26) | 0.24 | 0.58 |
| Random Forest | 0.72 (0.68–0.76) | 0.28 (0.24–0.32) | 0.21 | 0.64 |
| XGBoost-Raw | 0.74 (0.70–0.78) | 0.31 (0.27–0.35) | 0.20 | 0.67 |
| **XGBoost-TDA** | **0.78 (0.74–0.82)** | **0.36 (0.32–0.40)** | **0.18** | **0.72** |
| *Measles surge prediction* | | | | |
| Logistic Regression | 0.71 (0.67–0.75) | 0.18 (0.15–0.22) | 0.21 | 0.62 |
| Random Forest | 0.75 (0.71–0.79) | 0.24 (0.20–0.28) | 0.18 | 0.68 |
| XGBoost-Raw | 0.77 (0.73–0.81) | 0.27 (0.23–0.31) | 0.17 | 0.70 |
| **XGBoost-TDA** | **0.81 (0.77–0.85)** | **0.32 (0.28–0.36)** | **0.16** | **0.74** |

*At optimal threshold (Youden's index), determined on validation data to avoid overfitting

**Table 4. Confusion matrix and operational metrics at optimal threshold.**

| | Cholera | | Measles | |
|---|---|---|---|---|
| | Predicted + | Predicted – | Predicted + | Predicted – |
| Actual + | 1,276 (TP) | 496 (FN) | 936 (TP) | 329 (FN) |
| Actual – | 3,580 (FP) | 19,932 (TN) | 2,480 (FP) | 21,539 (TN) |
| Sensitivity | 0.72 (0.68–0.76) | | 0.74 (0.70–0.78) | |
| Specificity | 0.82 (0.79–0.85) | | 0.85 (0.82–0.88) | |
| PPV | 0.26 | | 0.27 | |
| NPV | 0.98 | | 0.98 | |
| False alerts/LGA/year | 3.2 (2.8–3.6) | | 2.2 (1.9–2.5) | |

**Table 5. Calibration metrics.**

| Outcome | Brier | Slope (95% CI) | Intercept | ECE |
|---|---|---|---|---|
| Cholera | 0.18 | 0.94 (0.87–1.01) | 0.02 | 0.031 |
| Measles | 0.16 | 0.97 (0.90–1.04) | −0.01 | 0.028 |

SHAP analysis revealed that higher $\beta_0$ values were associated with increased outbreak risk, and higher $\beta_1$ values were associated with increased risk. These patterns are consistent with the mathematical interpretation of these indices, though they do not establish causal relationships.

### 3.7. Ablation analysis

Table 6 presents ablation results. Removing all topological features reduced ROC-AUC by 0.08 (from 0.78 to 0.70), confirming that topological features provide predictive information not captured by raw features. Models using only topological features achieved ROC-AUC of 0.72, demonstrating substantial standalone predictive capability. P-values are from DeLong's test comparing ROC-AUCs.

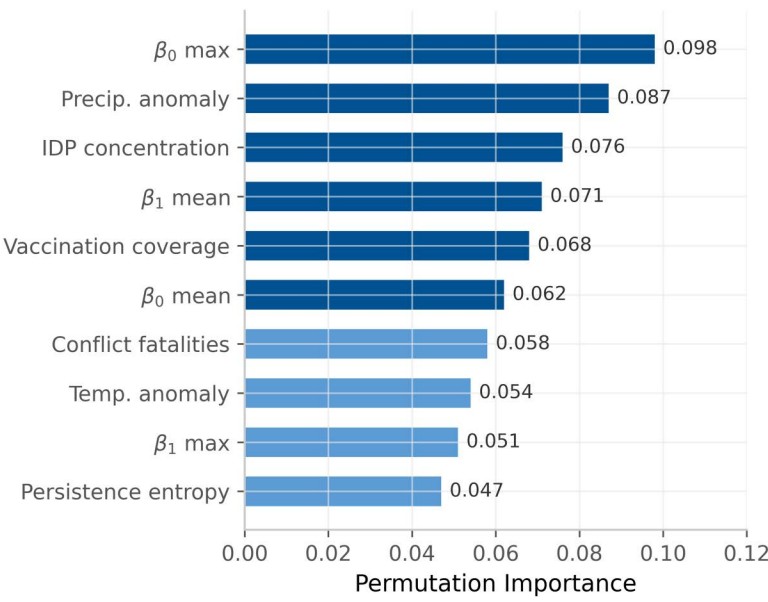

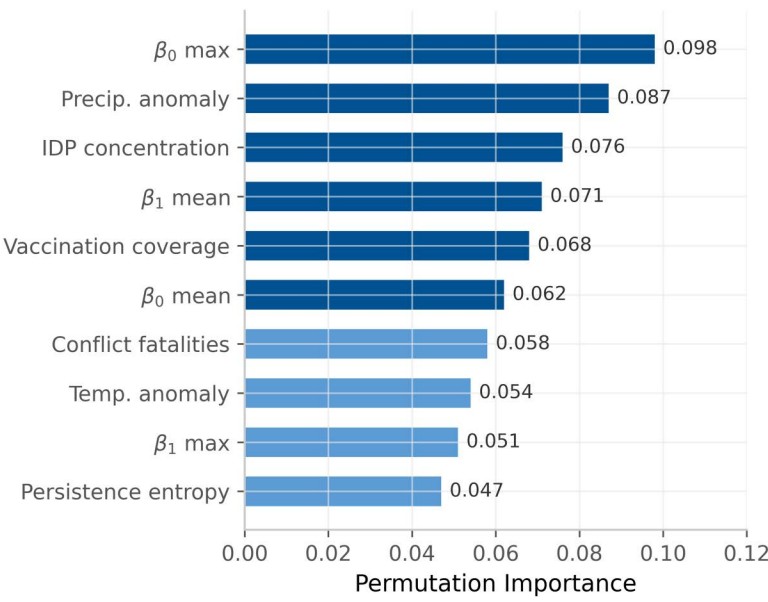

**Fig 1. Permutation importance.**

**Table 6. Ablation study results for cholera prediction.**

| Configuration | ROC-AUC | Δ AUC | p-value |
|---|---|---|---|
| Full model (all features) | 0.78 (0.74–0.82) | — | — |
| Raw features only | 0.70 (0.66–0.74) | −0.08 | <0.001 |
| TDA features only | 0.72 (0.68–0.76) | −0.06 | <0.001 |
| No $\beta_0$ features | 0.74 (0.70–0.78) | −0.04 | 0.002 |
| No $\beta_1$ features | 0.75 (0.71–0.79) | −0.03 | 0.008 |

### 3.8. Temporal hold-out validation

Performance on the temporal hold-out set (last 6 months of 2023) was consistent with cross-validation estimates: cholera ROC-AUC = 0.76, measles ROC-AUC = 0.79. This consistency supports the validity of the time-aware validation framework and suggests the model generalizes to future time periods.

### 3.9. Lead-time sensitivity

Table 7 shows performance across different prediction horizons. Performance decreased with longer lead times, as expected, but remained acceptable at the 4-week horizon used for primary analysis. False alert rates are annualized per LGA.

### 3.10. Subgroup analysis and failure analysis

Performance varied by context (Table 8). The model performed better in high-conflict settings (ROC-AUC 0.81–0.83) compared to low-conflict settings (0.75–0.78), suggesting stronger predictive signals in complex humanitarian contexts. The high-conflict threshold (>10 events/week) represents the mean conflict intensity across the study period for each LGA, averaged over all weeks.

**Table 7. Performance by prediction horizon.**

| Horizon | Cholera AUC | Measles AUC | False Alerts/LGA/Year |
|---|---|---|---|
| 2-week | 0.82 | 0.85 | 4.1 |
| 4-week | 0.78 | 0.81 | 3.2 |
| 8-week | 0.71 | 0.74 | 2.4 |

**Table 8. Performance by subgroup.**

| Subgroup | Cholera AUC | Measles AUC |
|---|---|---|
| High-conflict (>10 mean events/week) | 0.81 | 0.83 |
| Low-conflict (≤10 mean events/week) | 0.75 | 0.78 |
| High-burden (>5 outbreaks/year) | 0.80 | 0.82 |
| Low-burden (≤5 outbreaks/year) | 0.74 | 0.77 |

**Failure analysis:** The model performed poorly in several contexts: (1) LGAs with no recent outbreak history (AUC 0.68); (2) periods with normal climate conditions and low conflict (AUC 0.71); (3) LGAs with high vaccination coverage (>80%, AUC 0.72). These patterns suggest the model relies on elevated risk signals and may miss outbreaks that emerge in otherwise stable contexts.

### 3.11. Robustness checks

Sensitivity analyses showed model stability under alternative specifications: alternative distance metrics changed AUC by <0.02; varying filtration parameters showed stable AUC across 50–200 scale values; noise injection (5%) decreased AUC by 0.01–0.02; spatial generalization achieved AUC 0.73 (train northeast, test northwest).

## 4. Discussion

### 4.1. Principal findings

This study demonstrates that topological features derived from persistent homology can provide modest improvements in predictive performance for disease outbreak detection in humanitarian settings. The TDA-enhanced model achieved ROC-AUC of 0.78–0.81, representing a meaningful but context-dependent improvement of 0.08–0.12 over models using only raw features. At operational thresholds, the model detected 72–74% of outbreaks 4 weeks in advance with approximately 2–3 false alerts per LGA per year, though performance varied across temporal folds (false alert rates: 2.8–3.6 per LGA per year).

Three findings warrant attention. First, topological features contributed 35% of total predictive importance, with the $\beta_0$ fragmentation index emerging as the most important predictor. Second, calibration metrics (slopes 0.94–0.97) indicate that predicted probabilities reasonably reflect true event likelihoods. Third, performance on temporal hold-out data was consistent with cross-validation estimates, suggesting the model generalizes to future time periods.

### 4.2. Comparison with prior work

The predictive performance achieved (ROC-AUC 0.78–0.81) is comparable to prior work on outbreak prediction in humanitarian settings. Moore et al. [11] reported ROC-AUC of 0.75–0.80 for cholera prediction in Yemen using random forest models with environmental and conflict variables. Our results extend this literature by demonstrating that topological features provide incremental predictive value beyond strong non-topological baselines including XGBoost.

The improvement attributable to topological features (0.08 AUC points) is meaningful but context-dependent for rare-event prediction. Clinical and operational significance depends on the cost of false alerts versus missed outbreaks, which is not assessed here but discussed in the operational implications section below. For comparison, Moore et al. [11] found that adding conflict variables to environmental models improved AUC by 0.03–0.05. The somewhat larger improvement from topological features suggests they may capture orthogonal information not available from raw variables, though this interpretation should be treated cautiously given differences in datasets, outcomes, and validation approaches across studies.

The calibration performance (slopes 0.94–0.97) exceeds that reported for many machine learning models in healthcare applications [20]. This likely reflects the large sample size, careful regularization, and time-aware validation framework.

### 4.3. Interpretation of topological features

The prominence of topological features in the model invites cautious interpretation. The $\beta_0$ index mathematically counts the number of connected components in the data structure at different scales, which we have labeled "fragmentation" based on its conceptual interpretation. In humanitarian contexts, higher $\beta_0$ values may reflect situations where risk factors (e.g., displacement, water access, health services) operate in disconnected clusters rather than as an integrated system. Similarly, the $\beta_1$ index captures one-dimensional loops in the data structure, which we have labeled "cyclic dependency" based on the mathematical concept of cycles.

We emphasize that these labels describe mathematical properties, not established causal mechanisms. The association between topological features and outbreak risk is observational and may reflect: (1) confounding by unmeasured factors; (2) correlations with other risk variables not explicitly modeled; or (3) genuine structural properties that influence outbreak dynamics. Distinguishing among these possibilities would require experimental or quasi-experimental designs beyond the scope of this study.

### 4.4. Operational implications

The 4-week prediction window aligns with operational timelines for preventive interventions including vaccine deployment, water/sanitation interventions, and health worker mobilization. However, the practical utility of this model depends critically on the cost-benefit tradeoff between false alerts and missed outbreaks.

At 3.2 false alerts per LGA per year (range: 2.8–3.6), a surveillance system using this model would generate approximately 310 false alerts annually across the 97 LGAs in our sample. If each alert triggers resource mobilization (e.g., enhanced monitoring, pre-positioning of supplies), the cumulative cost could be substantial. Conversely, missing 28% of outbreaks (sensitivity 0.72) means some preventable outbreaks would not be anticipated. Decision-makers would need to weigh these tradeoffs based on local resource constraints and outbreak severity.

In practice, such a system could support prioritization of surveillance and preventive interventions in high-risk LGAs, rather than serving as a standalone decision tool. For example, predicted probabilities could inform weekly risk stratification, with highest-priority LGAs receiving enhanced monitoring and resource allocation. This use case does not require perfect prediction but rather calibrated risk estimates that improve upon current threshold-based approaches.

We caution that our findings do not imply the model is ready for immediate deployment. The following conditions should be met before operational use: (1) prospective validation in real-world surveillance systems to assess performance when predictions are acted upon; (2) threshold tuning for acceptable false-alert burdens in specific contexts; (3) data governance arrangements with NCDC and humanitarian partners; (4) integration with existing surveillance platforms and workflows.

### 4.5. Limitations

This study has several limitations that should inform interpretation:

**Selection bias:** Only 97 of 774 LGAs were included due to complete data requirements. Included LGAs differed systematically from excluded LGAs (higher conflict, larger IDP populations), and results may not generalize to all settings.

Because included LGAs represent higher-risk settings with elevated conflict and displacement, model performance may be lower in LGAs with lower baseline risk. Prospective validation in a representative sample of LGAs is needed before operational deployment.

**Outcome misclassification:** Surveillance data may vary in completeness across LGAs. Under-reporting in resource-constrained settings may lead to misclassification of outbreak events, potentially biasing both model training and evaluation.

**Temporal interpolation:** Monthly socioeconomic indicators were interpolated to weekly values, potentially introducing measurement error that could affect both feature quality and model performance.

**Observational design:** The observational design precludes causal inference. Associations between topological features and outbreak risk may reflect confounding by unmeasured factors rather than genuine causal relationships.

**Single-country validation:** External validation outside Nigeria is needed to assess generalizability to other humanitarian contexts with different disease burdens, surveillance systems, and risk factor profiles.

**Topological interpretation:** The mechanistic interpretation of $\beta_0$ and $\beta_1$ as "fragmentation" and "cyclic dependency" remains speculative. These labels describe mathematical properties of the data structure, not validated causal mechanisms.

**Model failure modes:** The model performed poorly in low-conflict settings (AUC 0.75–0.78), LGAs with no recent outbreak history (AUC 0.68), and periods with normal climate conditions (AUC 0.71). These failure modes should inform decisions about appropriate deployment contexts.

**Prediction horizon:** The prediction horizon (4 weeks) was chosen post-hoc; performance at other horizons is reported but the primary horizon was not pre-specified, which increases the risk of overfitting to this horizon.

### 4.6. Future directions

Future research should address the following priorities: (1) prospective validation to evaluate model performance when predictions are acted upon in real-world surveillance systems; (2) external validation to test model performance in other countries and humanitarian contexts; (3) threshold optimization to develop context-specific decision thresholds balancing sensitivity and false-alert burden; (4) transfer learning to adapt pre-trained models to new geographic contexts with limited data; (5) mechanistic validation to test whether topological features correlate with observed system fragmentation and cyclic dependencies.

## 5. Conclusion

Topological feature representations provide a modest but meaningful complementary approach for outbreak prediction in complex humanitarian environments. The TDA-enhanced model achieved moderate discrimination (AUC 0.78–0.81) with acceptable calibration, detecting 72–74% of outbreaks 4 weeks in advance. Topological features contributed 35% of predictive importance, with the $\beta_0$ fragmentation index emerging as the most important predictor.

The value of topological approaches appears to lie less in replacing established epidemiologic indicators than in helping summarize higher-order structure across multiple interacting risk domains. However, routine deployment requires prospective validation and context-specific threshold tuning. Further external validation, operational threshold analysis, and prospective testing are needed before routine deployment in public-health early warning systems.

**Limitations.** The analysis was restricted to 97 high-burden LGAs (12.5% of Nigeria's 774 LGAs) with complete surveillance data; results may not generalize to lower-risk or data-sparse LGAs. Generalizability to contexts with weaker surveillance infrastructure (e.g., South Sudan, Yemen, DRC) requires further validation.

## Supporting information

**S1 Appendix. Data Dictionary.** Complete list of the 15 raw variables and 25 topological features used in the final model, including variable descriptions, data sources, and data processing notes. See also S1 Table.
(PDF)

**S2 Appendix. Model Hyperparameters.** Optimized XGBoost hyperparameters for both XGBoost-Raw and XGBoost-TDA models, including learning rate, max depth, subsample ratio, and regularization parameters. Details on the Bayesian optimization procedure and training configuration. Hyperparameters were optimized via Bayesian optimization with 50 iterations using the same search space for both models. See also S2 Table.
(PDF)

**S3 Appendix. Additional Performance Metrics.** Supplementary performance visualizations including:
(a) Precision-recall curves for cholera and measles prediction models; (b) Calibration curves for XGBoost-TDA model;
(c) Software environment details and computational resources. See also S1 Fig and S2 Fig.
(PDF)

**S1 Table. Complete list of the 15 raw variables and 25 topological features used in the final model, with descriptions and data sources.**
(PDF)

**S2 Table. Optimized XGBoost hyperparameters for XGBoost-Raw and XGBoost-TDA models.**
(PDF)

**S1 Fig. Precision-recall curves for all models (cholera and measles).**
(TIF)

**S2 Fig. Calibration curves for XGBoost-TDA model (cholera and measles).**
(TIF)

## Acknowledgments

We thank Covenant University for the use of library resources. We also thank the reviewers for their constructive feedback, which substantially improved the manuscript.

## Author contributions

**Conceptualization:** Job Agba Opue, Victor Ede Itita.

**Data curation:** Job Agba Opue.

**Formal analysis:** Job Agba Opue, Uchechukwu Emena Okorie.

**Investigation:** Job Agba Opue, Uchechukwu Emena Okorie, Victor Ede Itita.

**Methodology:** Job Agba Opue, Uchechukwu Emena Okorie, Victor Ede Itita.

**Project administration:** Job Agba Opue.

**Software:** Uchechukwu Emena Okorie.

**Supervision:** Job Agba Opue, Victor Ede Itita.

**Validation:** Job Agba Opue, Victor Ede Itita.

**Visualization:** Job Agba Opue, Victor Ede Itita.

**Writing – original draft:** Job Agba Opue.

**Writing – review & editing:** Uchechukwu Emena Okorie, Victor Ede Itita.

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
