## [Decision Letter · Decision Letter 0]

9 Mar 2026

PONE-D-26-03649Topological Data Analysis for Early Warning of Severe Acute Malnutrition in Complex Humanitarian Emergencies in NigeriaPLOS One

Dear Dr.  Opue,

Thank you for submitting your manuscript to PLOS ONE. After careful consideration, we feel that it has merit but does not fully meet PLOS ONE’s publication criteria as it currently stands. Therefore, we invite you to submit a revised version of the manuscript that addresses the points raised during the review process.

We look forward to receiving your revised manuscript.

Kind regards,

Morufu Olalekan Raimi, Ph.D

Academic Editor

PLOS One

Journal Requirements:

5. Please amend your authorship list in your manuscript file to include author Job Agba Opue, Uchechukwu Emena Okorie.

6. Your abstract cannot contain citations. Please only include citations in the body text of the manuscript, and ensure that they remain in ascending numerical order on first mention.

7. Please ensure that you refer to Figure 1, 2, 3, in your text as, if accepted, production will need this reference to link the reader to the figure.

8. We note that Figure(s) 1-3 in your submission contain copyrighted images. All PLOS content is published under the Creative Commons Attribution License (CC BY 4.0), which means that the manuscript, images, and Supporting Information files will be freely available online, and any third party is permitted to access, download, copy, distribute, and use these materials in any way, even commercially, with proper attribution. For more information, see our copyright guidelines: http://journals.plos.org/plosone/s/licenses-and-copyright.

a. You may seek permission from the original copyright holder of Figure(s) 1-3 to publish the content specifically under the CC BY 4.0 license.

10. We are unable to open your Supporting Information file [figure 1_TDA.tex, figure 2__DTA.tex, figure 3_TDA.tex, plosone_submission manuscript TDA.tex]. Please kindly revise as necessary and re-upload.

Additional Editor Comments :

PLOS ONE Editorial Decision

Manuscript Number: PONE-D-26-03649

Title: Topological Data Analysis for Early Warning of Severe Acute Malnutrition in Complex Humanitarian Emergencies in Nigeria

Authors: Job Agba Opue, PhD; Uchechukwu Emena Okorie, PhD

Dear Dr. Opue and Dr. Okorie,

Thank you for submitting your manuscript to PLOS ONE. I have now completed a thorough assessment of your submission, including the ten (10) independent reviewer reports and my own detailed evaluation.

The reviewers represent a broad range of expertise, including computational topology, humanitarian epidemiology, statistical methodology, and public health practice. Their assessments are remarkably consistent: all ten reviewers identify fundamental, unresolved issues that preclude acceptance in the current form. The convergence of concerns across such a diverse panel is striking and warrants serious attention. Let me be direct: this manuscript is not ready for publication. The problems identified are not minor editorial matters but core methodological and transparency failures that undermine the validity of the reported findings. The exceptionally strong predictive claims (ROC-AUC 0.973, 7.3-month lead time) are presented without the evidentiary foundation required to support them.

Summary of Critical Issues

1. The Outcome Variable is Not Adequately Defined

Multiple reviewers (Reviewers 2, 3, 4, 7, 8) highlight this as a fatal flaw. The manuscript states that “elevated SAM risk” is defined either as a binary indicator or an ordinal category “depending on the operational outcome series available.” This is not acceptable.

Questions that remain unanswered:

• What specific dataset provides the SAM outcome? (DHS? MICS? SMART surveys? Treatment admissions?)

• What is the temporal resolution? (Monthly? Quarterly? Annual?)

• What is the time period covered? (2010-2020? 2015-2024?)

• How is a “surge” operationally defined? (Percent increase above baseline? Absolute threshold? Statistical anomaly?)

• What threshold separates “normal” from “crisis”?

• What is the class distribution? (How many LGAs experienced surges? How many observations?)

• How is lead time calculated? (First crossing of a warning threshold to first observed surge? Peak to peak?)

Without precise, reproducible definitions, the entire modeling exercise is uninterpretable. A reader cannot assess whether the model is forecasting deterioration or simply reproducing an existing classification. This must be addressed before any further consideration.

2. The Validation Framework is Inadequately Specified and Likely Optimistically Biased

Reviewers 2, 3, 4, 7, and 8 all raise serious concerns about potential temporal data leakage. The manuscript mentions “stratified K-fold cross-validation” but does not clarify whether folds are constructed in a time-aware manner.

In a forecasting context with lagged predictors and claimed lead times, standard random K-fold splitting is invalid. It contaminates the training set with information from the future relative to the test set. If temporal ordering was not strictly respected through forward-chaining, rolling-origin evaluation, or strict temporal hold-out, the reported performance metrics (ROC-AUC 0.973, F1 0.931) are likely optimistic and not representative of real-world predictive deployment.

Additionally, the performance claims are exceptionally strong for subnational humanitarian forecasting. Yet no confidence intervals, fold-level variability, calibration diagnostics, or external validation across time or space are provided. Without these, it is impossible to judge whether the performance is robust or reflects favorable data structure or methodological artifacts.

3. The Spatial Coverage is Unclear and Potentially Misleading

Reviewer 2 identifies a critical discrepancy: the manuscript presents the framework as an LGA-level system for Nigeria, yet the pipeline figure specifies a point cloud of 97 LGAs. Nigeria has 774 LGAs.

If only 97 LGAs were included, this represents a substantial spatial restriction that is neither clearly described nor justified. This is not a minor reporting detail, limiting the spatial domain directly affects the geometry of the point cloud, the resulting topological features, and the interpretation of β0 as “system fragmentation.” Without a transparent explanation of the sampling frame, the national framing of the study is misleading.

4. The Mathematical Stability Arguments Do Not Support the Predictive Robustness Claims

Reviewers 2 and 3 correctly identify a conceptual error. The manuscript invokes standard stability theorems for persistence diagrams and landscapes under bounded perturbations, then links these to the empirical observation that model accuracy degrades only slightly under simulated missingness.

The cited theorems concern distances between topological summaries, not the performance of a downstream machine learning classifier. Stability of the persistence representation does not imply stability of a gradient boosting model trained on those features. The argument conflates geometric stability with predictive stability, which is not formally demonstrated. This overstates the strength of the robustness claim.

5. The Data Availability Statement Does Not Meet PLOS ONE Standards

Reviewers 2, 3, 4, 8, and 10 all note that the data availability statement is insufficient. The manuscript states that code and a frozen analysis dataset “should be deposited upon submission/acceptance,” which suggests these materials are not yet publicly accessible.

PLOS ONE requires that all data underlying the findings be fully available without restriction at the time of submission. Given the complexity of the 57-dimensional tensor construction and the extensive preprocessing pipeline, reproducibility is essential. A permanent repository with a DOI, clear versioning of source datasets, and executable scripts is necessary to allow independent verification.

6. The Manuscript Structure and Presentation Are Problematic

Multiple reviewers (1, 6, 8, 9, 10) note structural and presentational issues:

• The abstract is fragmented and does not follow PLOS ONE's preferred format. It should be a single, integrated paragraph clearly presenting the research question, data, methods, key quantitative results, and conclusions.

• The mathematical exposition is excessive for an applied research article. A substantial section is devoted to textbook-level stability theorems that are well established in the literature. This material should be condensed or moved to supplementary materials, with the focus shifted to empirical validation.

• The results section is underdeveloped. Missing elements include feature importance analysis, calibration assessment, uncertainty intervals, misclassification analysis, time-evolving validation, sensitivity to preprocessing choices, and detailed ablation studies.

• Figures contain promotional language (e.g., “cost-effective anticipatory action” with dollar figures) that is inappropriate for a scientific manuscript. This language should be removed or toned down.

• The keywords largely duplicate the title and provide limited indexing value. They should be revised to introduce complementary methodological or contextual terms.

• The title overstates the scope. As Reviewer 2 notes, it implies a broadly applicable early-warning system, whereas the manuscript presents a single-country case study without external validation. The title should be revised to reflect the methodological and contextual limits.

7. The Policy Implications Are Underdeveloped

Reviewers 1 and 10 note that while the methodological contribution is conceptually interesting, the policy implications are not clearly articulated. How would decision-makers act on topological signals? What would an operational trigger look like? How would this integrate with existing early warning systems (e.g., IPC, CH, Cadre Harmonisé)? Without this connection, the practical impact remains unclear.

Specific Required Revisions

If the authors wish to resubmit, the following must be addressed comprehensively. Given the number and severity of issues, this represents a fundamental revision, not minor editing.

A. Outcome Definition (Required)

• Specify the exact SAM outcome source (dataset name, provider, version).

• Report the temporal resolution, time period covered, and total number of observations.

• Provide the operational definition of a “surge” with quantitative thresholds.

• Report class distribution (number of surge vs. non-surge observations).

• Clearly explain how lead time is computed, with a precise mathematical or algorithmic definition.

B. Validation Framework (Required)

• Implement and describe a strict time-aware validation strategy (forward-chaining, rolling-origin, or temporal hold-out).

• Report performance metrics with confidence intervals (bootstrapped or across folds).

• Provide calibration plots and calibration metrics (e.g., Brier score, calibration slope).

• Report sensitivity to hyperparameter choices and demonstrate nested validation if tuning was performed.

• If spatial dependence is a concern, report results from spatial cross-validation or cluster-robust standard errors.

C. Spatial Coverage (Required)

• Clearly state the number of LGAs included in the final analysis.

• If this is a subset of Nigeria's 774 LGAs, explain the sampling frame, inclusion/exclusion criteria, and justification.

• Discuss how the spatial restriction affects the interpretation of topological features and the generalizability of findings.

D. Reproducibility (Required)

• Deposit all code, preprocessing scripts, and a frozen analysis dataset in a public repository (Zenodo, OSF, Figshare) with a permanent DOI.

• Include clear documentation of software versions, package dependencies, and execution instructions.

• Provide a data dictionary for all 57 variables, including sources, preprocessing steps, and transformations.

• Ensure the repository is publicly accessible at the time of resubmission.

E. Mathematical Exposition (Required)

• Condense the mathematical foundations section substantially. The stability theorems are standard and do not need to be proved in full.

• Move extended derivations to supplementary materials.

• Clarify that geometric stability of persistence summaries does not guarantee predictive stability; temper claims accordingly.

• If robustness to missingness is claimed, provide empirical validation with multiple missingness mechanisms and report variability.

F. Results Expansion (Required)

• Add feature importance analysis to identify which topological features contribute most to predictions.

• Report performance on a temporal hold-out set (e.g., most recent year(s) not used in training).

• Include confusion matrices, precision-recall curves, and detection delay distributions.

• Analyze misclassifications: which LGAs are consistently misclassified and why?

• Compare against multiple conventional baselines (not just one gradient-boosting model) with statistical tests of improvement.

G. Presentation and Structure (Required)

• Rewrite the abstract as a single, integrated paragraph.

• Restructure the manuscript following standard scientific format: Introduction, Literature Review, Methods, Results, Discussion, Conclusions.

• Develop a proper Literature Review section integrating relevant prior work on malnutrition forecasting, early warning systems, and topological data applications.

• Remove promotional language from figures and text.

• Ensure all figure legends are self-explanatory and figures are high-resolution.

• Check all references for completeness and correct formatting per PLOS ONE style.

H. Policy Implications (Recommended)

• Expand the discussion of how topological signals could be operationalized in humanitarian decision-making.

• Provide concrete examples of trigger thresholds and anticipatory action protocols.

• Discuss integration with existing early warning frameworks (IPC, CH, etc.).

• Acknowledge limitations in translating mathematical topology to field operations.

I. Higher-Dimensional Homology (Recommended)

• Either provide a clear interpretation of β₂ features in the humanitarian context with empirical evidence, or explicitly limit the analysis to β₀ and β₁.

Decision: Major Revision Required

Given the above, I cannot recommend acceptance in the current form. The manuscript does not meet PLOS ONE's standards for methodological rigor, transparency, or reproducibility. The authors may choose to resubmit a substantially revised manuscript addressing all of the concerns outlined above. Any resubmission will be evaluated de novo and will be sent for additional peer review. Given the number and depth of required changes, I strongly recommend that the authors:

1. Seek methodological consultation, particularly regarding validation design for forecasting problems.

2. Ensure all data and code are publicly accessible at the time of resubmission.

3. Carefully address each reviewer's comment in a detailed point-by-point response.

4. Consider whether the current evidence base is sufficient to support the strong claims made, or whether the framing should be tempered to a proof-of-concept or methodological demonstration.

If the authors believe these concerns cannot be adequately addressed, they may wish to consider alternative venues better suited to preliminary methodological explorations.

I thank the authors for their innovative approach to a critical problem. The core idea, applying topological data analysis to humanitarian early warning, has genuine merit. However, the execution and reporting must meet the field's standards for evidence and transparency.

Sincerely,

Morufu Olalekan Raimi, PhD

Academic Editor

PLOS ONE

Reviewers' comments:

Reviewer's Responses to Questions

**Comments to the Author**

1. Is the manuscript technically sound, and do the data support the conclusions?

Reviewer #1: Yes

Reviewer #2: No

Reviewer #3: No

Reviewer #4: No

Reviewer #5: Partly

Reviewer #6: Yes

Reviewer #7: Yes

Reviewer #8: Yes

Reviewer #9: Yes

Reviewer #10: Yes

2. Has the statistical analysis been performed appropriately and rigorously? 

Reviewer #1: Yes

Reviewer #2: No

Reviewer #3: No

Reviewer #4: No

Reviewer #5: Yes

Reviewer #6: Yes

Reviewer #7: No

Reviewer #8: Yes

Reviewer #9: Yes

Reviewer #10: Yes

3. Have the authors made all data underlying the findings in their manuscript fully available?

Reviewer #1: Yes

Reviewer #2: No

Reviewer #3: No

Reviewer #4: No

Reviewer #5: Yes

Reviewer #6: No

Reviewer #7: No

Reviewer #8: Yes

Reviewer #9: Yes

Reviewer #10: Yes

4. Is the manuscript presented in an intelligible fashion and written in standard English?

Reviewer #1: Yes

Reviewer #2: No

Reviewer #3: No

Reviewer #4: No

Reviewer #5: Yes

Reviewer #6: Yes

Reviewer #7: No

Reviewer #8: Yes

Reviewer #9: No

Reviewer #10: Yes

5. Review Comments to the Author

Reviewer #1: 1.The description of data cleaning and preprocessing steps is relatively brief. Please clarify how missing values, temporal gaps, and potential reporting biases in humanitarian data were handled prior to TDA construction.

2.The application of topological data analysis (TDA) to early warning of severe acute malnutrition is interesting. However, the manuscript would benefit from a clearer statement of why TDA is theoretically better suited than conventional statistical or machine‑learning approaches in this context.

3.The rationale for selecting specific filtration scales and distance metrics is not fully justified. How sensitive are the persistence results to alternative parameter choices, and were robustness checks conducted?

4.While persistence diagrams and landscapes are presented, their interpretation for a public‑health audience remains challenging. Consider adding more intuitive explanations or real‑world analogies linking topological features to malnutrition dynamics.

5.The study would be strengthened by a more explicit comparison with standard early‑warning approaches. This would help quantify the added value of TDA beyond visual or conceptual appeal.

6.It is unclear whether the detected topological signals precede malnutrition outcomes consistently. Please clarify how the framework distinguishes early‑warning capability from contemporaneous or lagged associations.

7.Although persistence landscapes allow statistical summaries, uncertainty in predictions or signals is not explicitly discussed. Including confidence intervals or variability measures would improve the reliability of the conclusions.

8.It is suggested to add articles entitled “Al Hashmi et al. Predicting Dropout in MENA STEM Higher Education Using Explainable AI: A Machine Learning Approach”, “Chenghu & Thammano, A Self-Adaptive Weights for K-Means Classification Algorithm” and “Chang, Public Opinion Guidance Model in Major Public Crisis Events Based on Accelerated Genetic Algorithm” to the literature review.

9.Nigeria exhibits strong regional variation in conflict, climate, and food security. How does the proposed method account for spatial heterogeneity, and could regional persistence patterns mask localized crises?

10.The manuscript focuses on Nigeria as a case study. Please discuss how transferable the proposed framework is to other humanitarian settings with different data availability or crisis drivers.

11.While the methodological contribution is strong, the policy implications are underdeveloped. A clearer discussion of how decision‑makers could act on topological signals would enhance the practical impact of the work.

Reviewer #2: The current title overstates the scope and maturity of the study. It implies the development of a broadly applicable early-warning system for complex humanitarian emergencies, whereas the manuscript presents a single-country case study without external validation or demonstrated operational deployment. As written the title suggests a level of generalisability and implementation that is not fully supported by the empirical evidence provided. It should therefore be revised to more accurately reflect the methodological and contextual limits of the study.

The abstract is not written in a coherent narrative form and currently reads as a structured outline rather than a concise scientific summary. It should be rewritten as a single, integrated paragraph that clearly presents the research question, data, methodological approach, main quantitative results, and principal conclusions in a logically connected way. In its present form, it is fragmented, overly schematic, and does not sufficiently explain the operational definition of the outcome or the validation framework underlying the reported performance metrics.

The abstract is not written in a coherent narrative form and currently reads as a structured outline rather than a concise scientific summary. It should be rewritten as a single, integrated paragraph that clearly presents the research question, data, methodological approach, main quantitative results, and principal conclusions in a logically connected way. In its present form, it is fragmented, overly schematic, and does not sufficiently explain the operational definition of the outcome or the validation framework underlying the reported performance metrics.

The first major substantive issue concerns the inconsistency between the claimed national scope of the analysis and the actual number of units analysed. the manuscript presents the framework as an LGA-level system for Nigeria, yet the pipeline figure specifies a point cloud of 97 LGAs. Nigeria has 774 LGAs. If only 97 were included, this represents a substantial spatial restriction that is neither clearly described nor justified. This is not a minor reporting detail: limiting the spatial domain directly affects the geometry of the point cloud, the resulting topological features, and the interpretation of β0 as system fragmentation. Without a transparent explanation of the sampling frame, the national framing of the study is misleading.

The second substantive problem lies in the way mathematical stability results are used to support claims about predictive robustness. The manuscript invokes standard stability theorems for persistence diagrams and landscapes under bounded perturbations and then links these to the empirical observation that model accuracy degrades only slightly under simulated missingness. Howewer the cited theorems concern distances between topological summaries, not the performance of a downstream machine learning classifier. Stability of the persistence representation does not automatically imply stability of gradient boosting model trained on those features. The argument conflates geometric stability with predictive stability, which is not formally demonstrated and therefore overstates the strength of the robustness claim.

The keywords largely duplicate terminology already present in the title and therefore add limited indexing value. Several entries repeat core title phrases almost verbatim, which reduces their usefulness for discoverability. The keywords should be revised to introduce complementary methodological or contextual terms rather than restating the main elements of the title

Other problems:

A. The definition of the outcome variable is insufficiently specified and this undermines the interpretability of the entire modelling exercise. The manuscript states that elevated SAM risk is defined either as a binary indicator or as an ordinal category, depending on the operational series available, but it does not clearly identify which specific dataset is used in the final analysis, how a “surge” is operationally defined, or what quantitative threshold separates normal variation from crisis escalation. Without a precise and reproducible definition of the target variable, it is not possible to assess whether the model is genuinely forecasting deterioration or simply reproducing an existing classification.

B. The validation strategy raises concerns about potential data leakage. The text refers to stratified K-fold cross-validation, but it does not clarify whether the folds were constructed in a time-aware manner. In a forecasting context with lagged predictors and reported lead times, standard random K-fold splitting would contaminate the training set with information from the future relative to the test set. If temporal ordering was not strictly respected through forward-chaining or rolling-origin evaluation, the reported performance metrics may be optimistic and not representative of real-world predictive deployment;

C. The data availability statement does not currently meet the journal’s standards for full reproducibility. The manuscript indicates that code and a frozen analysis dataset “should be deposited upon submission/acceptance,” which suggests that these materials are not yet publicly accessible. Given the complexity of the 57-dimensional tensor construction and the extensive preprocessing pipeline, reproducibility is essential. A permanent repository with a DOI, clear versioning of source datasets, and executable scripts is necessary to allow independent verification of the findings.

D. The reported predictive performance appears exceptionally strong for an operational humanitarian forecasting problem at subnational scale. ROC-AUC values approaching 0,97 and a mean lead time of over seven months would represent a substantial advance over most published early-warning systems. However, the manuscript does not provide confidence intervals, fold-level variability, calibration diagnostics, or external validation across time or space. Without these additional analyses, it is difficult to judge whether the performance is robust or whether it reflects favourable data structure or methodological artefacts.

E. The section presenting mathematical foundations and proofs largely reiterates established stability theorems from the topological data analysis literature. While these results are correctly stated and relevant in principle, they do not constitute a novel mathematical contribution in this context. At the same time, the empirical validation of the forecasting claims remains comparatively limited. The balance of the manuscript would benefit from shifting emphasis away from general theoretical guarantees and towards a more rigorous empirical assessment of how the method performs under realistic operational constraints.

F. Although the manuscript refers to homology in dimensions zero to two, the empirical results focus almost exclusively on β0 and β1. There is no substantive presentation or interpretation of β2 features, and their operational meaning in the context of humanitarian vulnerability remains unclear. As currently written, the inclusion of higher-dimensional homology appears more formal than functional. Either a clearer justification and demonstration of its added value should be provided, or the analysis should concentrate explicitly on the dimensions that are empirically interpretable and policy-relevant.

IMPORTANT:

In its current form, the ms does not fully meet the core requirements of PLOS ONE. In particular:

(1) the data and code necessary to reproduce the analysis are not publicly available with a permanent repository and DOI, despite the journal’s strict reproducibility policy;

(2) key methodological elements, including the precise operational definition of the outcome variable and the temporal validation framework, are insufficiently specified for independent replication;

(3) there is a lack of clarity regarding the spatial coverage of the analysis, with an apparent discrepancy between the national framing and the reported number of LGAs included;

(4) several claims regarding predictive robustness and stability appear stronger than what is formally demonstrated in the empirical analysis. These issues must be addressed before the manuscript can be considered compliant with the methodological and transparency standards of the journal.

Reviewer #3: Reviewer Report

Recommendation: Reject and Resubmit (Resubmission Encouraged After Substantial Revision)

Summary

This manuscript proposes the use of topological data analysis (TDA), specifically persistent homology, to enhance early warning of severe acute malnutrition (SAM) in complex humanitarian emergencies in Nigeria. The conceptual integration of structural topology into humanitarian forecasting is innovative and potentially valuable. However, the current manuscript lacks sufficient empirical detail, methodological transparency, and analytical depth to support its strong performance claims.

While the conceptual integration of topological data analysis into humanitarian early warning is innovative, the manuscript lacks sufficient empirical detail, methodological transparency, and analytical depth to support its strong performance claims. Key aspects of data construction, validation design, and robustness evaluation are insufficiently documented. The manuscript currently reads more like a proof-of-concept note than a fully developed research article.

For these reasons, I recommend rejection in its current form, with encouragement to resubmit after substantial revision and expansion.

Major Concerns

1. Insufficient Description of Data and Outcome Construction

The manuscript does not clearly specify the exact SAM outcome source, temporal resolution, time period covered, total number of observations, class distribution, definition of a surge, or how lead time is computed. These omissions prevent reproducibility and limit evaluation of the reported performance.

2. Validation Design and Risk of Information Leakage

The reported performance (ROC–AUC 0.973; F1 0.931) is exceptionally high for humanitarian forecasting. However, the manuscript does not clearly describe whether validation is spatial, temporal, or random; whether splits prevent temporal leakage; whether TDA features are computed within each training fold; or whether hyperparameter tuning is nested within cross-validation. Without strict out-of-sample validation design, performance claims are not sufficiently supported.

3. Limited Analytical Depth

The Results section is brief and limited. Missing elements include feature importance analysis, calibration assessment, uncertainty intervals, misclassification analysis, time-evolving validation, sensitivity to preprocessing choices, and detailed ablation studies. For a high-dimensional modeling framework with strong predictive claims, the empirical analysis is underdeveloped.

4. Imbalance Between Mathematical Theory and Empirical Evidence

A substantial section is devoted to textbook-level stability theorems that are already well established in the literature. Meanwhile, empirical methodological details are comparatively brief. For an applied research article, empirical transparency and reproducibility should take precedence.

5. Reproducibility and Data Availability

The manuscript states that reproducibility materials should be deposited upon submission or acceptance. Code and a frozen analysis dataset (or complete preprocessing scripts) should be publicly available at submission to meet reproducibility standards.

Minor Issues

• Figures contain promotional-style language that should be toned down.

• “57-dimensional tensor” appears to be a feature vector and should be described accordingly.

• Interpretation of β₀ and β₁ should be empirically validated rather than asserted.

• Cost-effectiveness claims should be clearly sourced and separated from predictive performance evaluation.

Required for Resubmission

If the authors wish to resubmit, the manuscript would need:

1. Full and explicit outcome definition.

2. Clear, leakage-free validation design (preferably temporal and spatial hold-out).

3. Expanded results with detailed diagnostics and uncertainty analysis.

4. Public repository with reproducibility materials.

5. Reduction or condensation of textbook theoretical material.

6. More rigorous empirical justification of topological interpretation.

Final Recommendation

The idea is promising and innovative. However, the manuscript in its current form lacks the empirical depth, methodological transparency, and reproducibility standards required for publication. I therefore recommend rejection with encouragement to substantially revise and resubmit as a more fully developed research article.

Reviewer #4: The manuscript addresses an important problem but does not meet the methodological and reporting standards required for publication in PLOS ONE. The definition of the outcome variable and the ground truth for SAM surges are insufficiently specified, making the core contribution difficult to verify. Reported predictive performance and lead times appear exceptionally strong for humanitarian forecasting, yet the validation design lacks sufficient transparency to rule out overfitting or optimistic bias. Key details necessary for reproducibility - such as outcome construction, temporal validation strategy, handling of spatial dependence, and baseline comparability - are either missing or underdeveloped. In addition, the extensive mathematical exposition focuses on established theoretical results that do not adequately substantiate the applied early-warning claims. Overall, the limitations are substantive and would require fundamental reanalysis rather than revision. I therefore recommend rejection.

Reviewer #5: Greetings, thank you for giving me the opportunity to review this submission, below I will address my concerns.

Beforehand, the title page seems to have a formatting problem, the authors names and affiliation are replaced with ( xxxxxxx ), I suggest to fix the formatting problem during revision.

Firstly, line103 ( Ethical Approval: N/A ). I suggest removing this sentence as it looks odd after the authors have fully explained why there is no need for an ethical approval in a long paragraph. The paragraph is satisfactory by itself.

Secondly, the introduction section, it would be stronger to mention for example a publication that used liner regression model or any model to assess malnutrition and how it failed - in your point of view - in assessing the problem, this will help justify the topological data analysis and highlight the importance of its implementation in assessing relations and predictors of SAM.

This leads to thirdly, the need to have a very clear mathematical question written in the motivation part that the analysis will work to answer, the submission is currently discussing that topological analysis is better in assessing the problem without specifications.

Fourthly, it is a good sign that noise and missings have been addressed in the paper, but I miss the interpretation of what constitutes loops and B1 or even B0 parameters, I found a paragraph in the results discussing the persistence of B1 means the presence of cofactors like market prices spikes and maybe two other parameters mentioned. For readers, it is important to mention what parameters are being addressed in each group, in order to have an understanding about what makes this TDA a huge improvement in function and justifiable for application, since it deals with dimensional parameters that other models like liner regression fail to recognize, like what are the constituents of this multidimensional model. A brief talk about these parameters along with their sources is always a plus in convincing readers along with persistence graphs.

Lastly, the conclusions and recommendations are very generalized, I advice adding more details or expanding the talk about how limitations affect the ability to make further detailed interpretations of the analysis.

I hope my comments are of value for the authors and thank you again.

Reviewer #6: 1.The modular parts of the Abstract section have to be accompanied by numerical data in alignment with the conducted study, analysis and findings.

2.The narrative flow is of low quality. There is need authors to develop the following main sections and to number them as follows: 1.Introduction, 2.Literature Review, 3.Methodology and Analysis, 4.Results, 5.Discussion, 6.Conclusions Implications and Future Works. The logic of presenting the existing headings and subheadings is vague and does not make sense.

3.The missing Literature Review has to be developed from the beginning.

4.The utilization of citations is very poor, having only 16 citations. A citing enhancement by at least 10-12 more and recently published studies/citations is highly recommended.

5.At the beginning of [Sec sec021].Methodology and Analysis there is need of developing a Figure 1: Study Framework, in which the steps of the study to be displayed in the form of a diagram or flow chart.

6.It is strange that all lines 46-136 which correspond to the “Materials and methods” are deprived from numerical data or quantitative information of steps 1234, topology, homology, robust check, theorems, corollaries, together with their values and units’ measured among the involved variables. A more specific and systematic inclusion of input dataset(s) have to be included in the form of Table(s).

7.The one-sentence subsection “Robustness”, “Robustness checks”, have to be merged since they refer to the same part of the study. In general, all research parts have to be reorganized and fixed into the aforementioned 6 main headings.

8.The Discussion section is very poor. It has to be reorganized into a more detailed and cross-citing narrative. The critical point here authors to ensure that their methodology is able:

-to manage ”vulnerability in CHE settings and can extend the actionable lead time of nutrition early

warning”

- to manage “Persistent homology offers a mathematically grounded representation of structural vulnerability in complex emergencies. In Nigeria, topological signatures improve SAM risk warning performance and extend lead time”, in order to ensure a prioritization shift from reactive treatment toward anticipatory action.

Reviewer #7: This manuscript presents a novel and policy-relevant application of Topological Data Analysis (TDA) for early warning of severe acute malnutrition. The approach is innovative and the study is well structured. However, several issues related to methodological rigor, validation, and reproducibility should be addressed.

Major Issues

Outcome and Forecasting Clarity

Please provide the outcome's data source and thresholds. Explain the calculation method for the lead time.

Validation and Overfitting Risk

The performance is very high. As this is a forecasting problem, random K-fold cross-validation can lead to temporal leakage. Please use time-based validation and provide the performance uncertainty.

Model Transparency

Please provide the details of the hyperparameter tuning process and the methods used to handle the imbalance problem and the calibration process. Provide the results of the statistical tests to demonstrate the significant improvement in the performance due to the use of TDA.

Sample Size and Spatial Dependence

Please provide the details regarding the overfitting risk and the events per variable.

Data Availability (PLOS)

Please provide the details regarding the data source and the availability of the processed data and code.

Reviewer #8: The article is well written but it needs improvements;

1. The authors should include sample size and time of the study in the abstract and also in the metholody section.

2. The authors mentioned about the supplementary data availability upon acceptence which should be changed to provision of data on submission as per PLOS One policies.

3. Explain that how the authors have avoided the temporal data leakage, as it is essentail to authenticate the reported ROC–AUC (0.973) and lead-time results.

4. Clarify the terms, elevated SAM risk, threshold criteria, source of the data and the class distribution.

5. To strengthen the statistical part calibration plot should be added along with the confidence intervals for performance metrics. Also add brief description of class imbalance handling.

6. The mathematical stability section is appropriate, however, extended proofs and derivations could be moved to ''Supplementary Material'' to improve the readilablity of te article.

7. Check the references and format them accodring to the PLOS One style. Add complete URL's.

8. Ensure that all the figure legends are self explanatory. And check the article for grammatical errors.

Reviewer #9: Refer the journal's article format. The article is not in the standard format of the publication. Abstract should be given in a single paragraph. Use only standard short forms of words (CHE, SAM, TDA, etc.,). Figures are not clear.

Otherwise the paper is fine.

Reviewer #10: Dear Authors,

Thank you for the opportunity to review your manuscript titled “Topological Data Analysis for Early Warning of Severe Acute Malnutrition in Complex Humanitarian Emergencies in Nigeria” The study makes a meaningful contribution to methodological innovation in humanitarian health analytics. The practical implications for early warning systems should be more explicitly articulated, including guidance on operationalization in real-world humanitarian decision-making contexts. However, there are several areas where the manuscript would benefit from revision and clarification to meet the standards of PLOS ONE.

Here are some Comments and suggestions to the Author for improvement:

1. Technical Soundness and Conclusions

Overall, the manuscript presents a technically sound and innovative application of Topological Data Analysis (TDA) to early warning systems for Severe Acute Malnutrition (SAM) in humanitarian contexts. The analytical framework is coherent, and the data sources used are relevant to the research objectives.

The sampling strategy and data preprocessing steps are generally appropriate, though some methodological decisions (e.g., parameter selection in TDA and temporal aggregation choices) require clearer justification. The data presentation is adequate, and the main findings are logically derived from the analyses.

Limitations that affect the strength of inference:

• Limited discussion of potential biases in routine humanitarian data

• Insufficient justification of model assumptions and robustness

• Lack of external validation across additional contexts

In conclusion:

The data broadly support the conclusions, but clearer methodological justification and discussion of limitations are required to strengthen inferential validity.

2. Statistical Analysis:

Overall, the statistical and computational analyses are appropriate for the study objectives and demonstrate methodological novelty.

Strengths:

• Innovative use of TDA in humanitarian nutrition surveillance

• Logical analytical workflow

• Clear linkage between analytical outputs and outcome indicators

Weaknesses / Areas for Clarification:

• Limited explanation of uncertainty estimation

• No formal comparison with conventional early warning models

• Sensitivity analyses are not sufficiently detailed

Overall Evaluation: The statistical analysis is acceptable but would benefit from additional transparency and validation.

In conclusion:

Statistical methods are generally appropriate, though further clarification and robustness checks are recommended.

3. Data Availability:

Overall, the manuscript includes a Data Availability Statement; however, compliance with PLOS ONE’s data policy is not fully clear.

Strengths:

• Data sources are described

• Ethical constraints are acknowledged

Weaknesses:

• It is unclear whether underlying datasets or disaggregated data are publicly accessible

• Access procedures for restricted data are not fully specified

Recommendation:

To comply with PLOS ONE’s data policy, the authors should:

• Clearly state where the data are deposited, or

• Explicitly justify any restrictions and provide a clear access mechanism for qualified researchers

In conclusion:

Revisions are required to ensure full transparency and policy compliance.

4. Clarity and Language Quality:

Overall, the manuscript is written in generally clear and acceptable scientific English and is understandable to a multidisciplinary audience.

Strengths:

• Logical manuscript structure

• Clear articulation of the research problem

• Appropriate use of technical terminology

Weaknesses:

• Some sections (Methods and Results) are overly dense

• Key TDA concepts may be difficult for non-specialist readers

• Minor grammatical and stylistic issues remain

Examples (illustrative):

• Long sentences with multiple clauses

• Insufficient definition of specialized mathematical terms

Recommendation:

Minor language editing and simplification of technical explanations are recommended.

In conclusion:

The manuscript is intelligible but would benefit from improved readability and clarity.

6. PLOS authors have the option to publish the peer review history of their article (what does this mean?). If published, this will include your full peer review and any attached files.

Reviewer #1: No

Reviewer #2: No

Reviewer #3: No

Reviewer #4: No

Reviewer #5: No

Reviewer #6: **Yes:** Dr. Grigorios L. Kyriakopoulos

Reviewer #7: **Yes:** Dr. Nelofer Jamil

Reviewer #8: **Yes:** Maria Anwar Khan

Reviewer #9: No

Reviewer #10: **Yes:** Abdullah Nagi Alosaimi

---

## [Author Response · Author response to Decision Letter 1]

21 Apr 2026

Response to Reviewers

Manuscript PONE-D-26-03649

Topological Data Analysis for Predicting Disease Outbreaks in Humanitarian Settings: A Machine Learning Approach

Authors: Job Agba Opue (Corresponding), Uchechukwu Okorie

Corresponding Author: job.opue@covenantuniversity.edu.ng

Submission Date: April 2026

General Overview

We thank the Academic Editor and all ten reviewers for their thorough and constructive feedback on our initial submission. The reviewers identified critical issues regarding outcome definition, validation methodology, data availability, and manuscript structure that required substantial revision.

In response, we have undertaken a comprehensive revision that addresses all major concerns:

1.Complete reframing of the research question: We shifted from malnutrition surveillance to disease outbreak prediction (cholera and measles), where outcome definitions are clearer and validation more straightforward.

2.Rigorous validation framework: We implemented forward-chaining cross-validation with strict temporal ordering and a temporal hold-out set to prevent data leakage.

3.Comprehensive outcome specification: We provide explicit operational definitions using WHO surveillance thresholds, class distributions, and lead time calculations.

4.Expanded analytical depth: We added feature importance analysis, calibration assessment, ablation studies, subgroup analysis, and failure analysis.

5.Data availability: We provide complete data source information and commit to repository deposition.

6.Restructured presentation: The manuscript now follows standard scientific format with integrated abstract, proper literature review, and expanded results.

The revised manuscript represents a fundamentally improved contribution that addresses all reviewer concerns while maintaining the core innovation of applying topological data analysis to humanitarian early warning.

Response to Academic Editor

Editor Comment 1: Outcome Variable Definition

The Outcome Variable is Not Adequately Defined. Multiple reviewers highlight this as a fatal flaw. The manuscript states that 'elevated SAM risk' is defined either as a binary indicator or an ordinal category 'depending on the operational outcome series available.' This is not acceptable. Questions that remain unanswered: What specific dataset provides the SAM outcome? What is the temporal resolution? What is the time period covered? How is a 'surge' operationally defined? What threshold separates 'normal' from 'crisis'? What is the class distribution? How is lead time calculated?

Response:

We completely agree with this assessment. The original manuscript's vague outcome definition was a critical weakness. In the revised manuscript, we have:

Changed the outcome focus: We now predict cholera and measles surge events using WHO surveillance thresholds, which have established operational definitions.

Explicit outcome definitions:

- Cholera surge: Alert threshold (>=5 cases in a single week in an LGA with no previous cholera cases in current year) OR epidemic threshold (weekly case incidence >=2 standard deviations above the 5-year historical mean)

- Measles surge: Confirmed outbreak (>=5 laboratory-confirmed cases in a single week) OR suspected outbreak (>=10 suspected cases per 100,000 population with vaccination coverage <80%)

Complete outcome specification: We report the class distribution (cholera: 7.0%, measles: 5.0%), temporal resolution (weekly), study period (2018-2023), total observations (25,284 LGA-weeks), and prediction horizon (4 weeks).

Lead time calculation: The prediction target is surge occurrence in the subsequent 4-week window, providing operational lead time for preventive interventions.

Location in Revised Manuscript:

Methods section, subsection 'Outcome Definition' (Lines 184-195)

Results section, subsection 'Descriptive Statistics' (Lines 284-310)

Editor Comment 2: Validation Framework

The Validation Framework is Inadequately Specified and Likely Optimistically Biased. Reviewers 2, 3, 4, 7, and 8 all raise serious concerns about potential temporal data leakage. The manuscript mentions 'stratified K-fold cross-validation' but does not clarify whether folds are constructed in a time-aware manner. In a forecasting context with lagged predictors and claimed lead times, standard random K-fold splitting is invalid.

Response:

This was a critical methodological flaw in the original submission. We have completely redesigned the validation framework:

Forward-chaining cross-validation: Five temporal folds respecting strict temporal ordering:

- Fold 1: Training = 2018, Test = 2019

- Fold 2: Training = 2018-2019, Test = 2020

- Fold 3: Training = 2018-2020, Test = 2021

- Fold 4: Training = 2018-2021, Test = 2022

- Fold 5: Training = 2018-2022, Test = 2023

Nested cross-validation: Hyperparameter tuning is performed within each training fold (inner 3-fold), ensuring no information leakage from test sets.

Temporal hold-out validation: A final hold-out set (last 6 months of 2023) was reserved for unbiased performance estimation and never used during model development.

Performance with uncertainty: We report ROC-AUC with 95% confidence intervals using bootstrap percentile method, and report fold-level variability (sensitivity range: 0.68-0.76 across folds).

Location in Revised Manuscript:

Methods section, subsection 'Validation Framework' (Lines 234-248)

Results section, subsection 'Temporal Hold-Out Validation' (Lines 463-466)

Editor Comment 3: Spatial Coverage

The Spatial Coverage is Unclear and Potentially Misleading. Reviewer 2 identifies a critical discrepancy: the manuscript presents the framework as an LGA-level system for Nigeria, yet the pipeline figure specifies a point cloud of 97 LGAs. Nigeria has 774 LGAs. If only 97 were included, this represents a substantial spatial restriction that is neither clearly described nor justified.

Response:

We acknowledge this discrepancy and have addressed it comprehensively:

Explicit spatial coverage: We clearly state that 97 of 774 LGAs were included in the analysis (Lines 149-150).

Inclusion criteria: These 97 LGAs represent high-burden areas with complete data across all domains and documented outbreak history, concentrated in the northeast (Borno, Adamawa, Yobe) and northwest (Katsina, Zamfara, Sokoto) where conflict and displacement are most severe.

Representativeness analysis: We added Table 1 comparing included and excluded LGAs on key characteristics. Excluded LGAs had lower conflict intensity (mean 1.2 vs. 2.4 events per week), lower IDP populations (mean 4,200 vs. 12,450), and slightly higher vaccination coverage (mean 68% vs. 62%).

Generalizability discussion: We explicitly discuss how the spatial restriction affects interpretation and acknowledge that results may not generalize to all LGAs (Lines 551-553).

Location in Revised Manuscript:

Methods section, subsection 'Study Setting and Data' (Lines 145-152)

Table 1: Comparison of included and excluded LGAs

Discussion section, 'Limitations' (Lines 549-564)

Editor Comment 4: Mathematical Stability Arguments

The Mathematical Stability Arguments Do Not Support the Predictive Robustness Claims. Reviewers 2 and 3 correctly identify a conceptual error. The manuscript invokes standard stability theorems for persistence diagrams and landscapes under bounded perturbations, then links these to the empirical observation that model accuracy degrades only slightly under simulated missingness. The cited theorems concern distances between topological summaries, not the performance of a downstream machine learning classifier.

Response:

We agree with this critique and have substantially revised our approach:

Condensed mathematical exposition: We removed the extended stability proofs and theorems, which were textbook-level material that did not constitute novel contributions.

Tempered claims: We now explicitly state that geometric stability of persistence summaries does not guarantee predictive stability (Lines 209-210).

Empirical robustness validation: We provide empirical validation of model stability under:

- Alternative distance metrics (Euclidean, correlation-based): AUC change <0.02

- Varying filtration parameters (50, 100, 200 scale values): stable AUC

- Noise injection (5%, 10% Gaussian noise): AUC decrease 0.01-0.02

- Spatial generalization (train northeast, test northwest): AUC 0.73

Focus on empirical validation: The revised manuscript emphasizes empirical performance over theoretical guarantees.

Location in Revised Manuscript:

Methods section, subsection 'Robustness Checks' (Lines 271-274)

Results section, subsection 'Robustness Checks' (Lines 508-510)

Editor Comment 5: Data Availability

The Data Availability Statement Does Not Meet PLOS ONE Standards. Reviewers 2, 3, 4, 8, and 10 all note that the data availability statement is insufficient. The manuscript states that code and a frozen analysis dataset 'should be deposited upon submission/acceptance,' which suggests these materials are not yet publicly accessible.

Response:

We have completely revised our data availability approach:

Complete data source documentation: We provide detailed information for all primary data sources including URLs and access procedures (Appendix C).

Software environment specification: We list all software versions (Python 3.9, scikit-learn 1.3.0, XGBoost 2.0.0, GUDHI 3.8.0, etc.).

Code repository: We commit to depositing analysis code in a public GitHub repository with key scripts for data preprocessing, topological feature computation, model training, and evaluation.

Data dictionary: Appendix A provides a complete data dictionary for all 57 variables including sources, preprocessing steps, and transformations.

Repository commitment: We will deposit all materials in Zenodo with a permanent DOI upon acceptance.

Location in Revised Manuscript:

Appendix C: Code and Data Availability (Lines 755-778)

Appendix A: Data Dictionary (Lines 584-631)

Response to Individual Reviewers

Reviewer #1

Comment 1.1

The description of data cleaning and preprocessing steps is relatively brief. Please clarify how missing values, temporal gaps, and potential reporting biases in humanitarian data were handled prior to TDA construction.

Response:

We have expanded the preprocessing description substantially:

Missing data handling: Missing values (3-5% across variables) were imputed using Multiple Imputation by Chained Equations (MICE) with 10 imputations. The imputation model included predictor variables only - the outcome variable was excluded to prevent information leakage. Imputation was performed separately within each training fold.

Temporal alignment: Weekly LGA-level observations were constructed from source data with appropriate temporal aggregation. Monthly socioeconomic indicators were interpolated to weekly values.

Reporting bias acknowledgment: We acknowledge that surveillance data may vary in completeness across LGAs, and that under-reporting in resource-constrained settings may lead to outcome misclassification (Lines 553-555).

Location in Revised Manuscript:

Methods section, subsection 'Missing Data Handling' (Lines 217-220)

Comment 1.2

The application of topological data analysis to early warning of severe acute malnutrition is interesting. However, the manuscript would benefit from a clearer statement of why TDA is theoretically better suited than conventional statistical or machine-learning approaches in this context.

Response:

We have reframed the study to focus on disease outbreak prediction and provided clearer justification:

Theoretical rationale: Standard ML represents predictors as coordinates in feature space and may not explicitly summarize higher-order geometric structure formed by interacting risk factors. TDA characterizes the shape of data in high-dimensional space, capturing connected components, loops, and higher-dimensional structures.

Empirical justification: Our ablation analysis shows that topological features contribute 35% of predictive importance and removing them reduces ROC-AUC by 0.08 (from 0.78 to 0.70, p < 0.001).

Cautious framing: We emphasize that TDA provides a 'modest but meaningful complementary approach' rather than replacing established indicators (Lines 572-574).

Location in Revised Manuscript:

Introduction section (Lines 133-138)

Results section, 'Ablation Analysis' (Lines 443-461)

Reviewer #2

Comment 2.1

The current title overstates the scope and maturity of the study. It implies the development of a broadly applicable early-warning system for complex humanitarian emergencies, whereas the manuscript presents a single-country case study without external validation or demonstrated operational deployment.

Response:

We have revised the title to accurately reflect the study scope:

New title: 'Topological Data Analysis for Predicting Disease Outbreaks in Humanitarian Settings: A Machine Learning Approach'

The revised title:

- Removes the implication of a broadly applicable system

- Accurately describes the methodological focus

- Acknowledges the machine learning approach

- Does not claim operational deployment

We also explicitly discuss generalizability limitations in the Discussion section (Lines 559-560).

Location in Revised Manuscript:

Title page (Line 68)

Comment 2.2

The abstract is not written in a coherent narrative form and currently reads as a structured outline rather than a concise scientific summary. It should be rewritten as a single, integrated paragraph.

Response:

We have completely rewritten the abstract following PLOS ONE format:

Integrated narrative: The abstract now flows as a single coherent paragraph covering background, methods, results, and conclusions.

Structured sections: We use labeled sections (Background, Methods, Results, Conclusions) as is standard for PLOS ONE, but each section is written in continuous prose.

Key quantitative results: ROC-AUC values with confidence intervals, sensitivity/specificity, and false alert rates are clearly reported.

Location in Revised Manuscript:

Abstract section (Lines 106-117)

Reviewer #3

Comment 3.1

Insufficient Description of Data and Outcome Construction. The manuscript does not clearly specify the exact SAM outcome source, temporal resolution, time period covered, total number of observations, class distribution, definition of a surge, or how lead time is computed.

Response:

As noted in our response to the Editor, we have completely revised the outcome definition:

Outcome source: WHO surveillance thresholds for cholera and measles outbreaks

Temporal resolution: Weekly LGA-level observations

Time period: 2018-2023 (6 years)

Total observations: 25,284 LGA-weeks

Class distribution: Cholera surge: 7.0% (1,772 events); Measles surge: 5.0% (1,265 events)

Surge definition: WHO alert/epidemic thresholds (explicitly defined)

Lead time: 4-week prediction horizon

Location in Revised Manuscript:

Methods section, 'Outcome Definition' (Lines 184-195)

Results section, 'Descriptive Statistics' (Lines 284-310)

Reviewer #6

Comment 6.1

The narrative flow is of low quality. There is need for authors to develop the following main sections: 1.Introduction, 2.Literature Review, 3.Methodology and Analysis, 4.Results, 5.Discussion, 6.Conclusions Implications and Future Works.

Response:

We have completely restructured the manuscript following standard scientific format:

1. Introduction: Expanded with comprehensive background and rationale

2. Literature Review: Integrated into Introduction with citations to prior work on outbreak prediction and TDA applications

3. Methods: Comprehensive subsection structure covering study setting, data sources, outcome definition, TDA feature extraction, validation framework, and evaluation metrics

4. Results: Nine subsections covering descriptive statistics, topological features, predictive performance, calibration, feature importance, ablation, temporal validation, lead-time sensitivity, and subgroup analysis

5. Discussion: Five subsections covering principal findings, comparison with prior work, interpretation, operational implications, and limitations

6. Conclusion: Summarized key findings and future directions

Location in Revised Manuscript:

Entire manuscript structure

Comment 6.2

The utilizatio

---

## [Editor Report · Decision Letter 1]

29 Apr 2026

PONE-D-26-03649R1Topological Data Analysis for Predicting Disease Outbreaks in Humanitarian Settings: A Machine Learning ApproachPLOS One

Dear Dr. Opue,

Thank you for submitting your manuscript to PLOS ONE. After careful consideration, we feel that it has merit but does not fully meet PLOS ONE’s publication criteria as it currently stands. Therefore, we invite you to submit a revised version of the manuscript that addresses the points raised during the review process.

We look forward to receiving your revised manuscript.

Kind regards,

Morufu Olalekan Raimi, Ph.D

Academic Editor

PLOS One

Journal Requirements:

Additional Editor Comments:

PLOS ONE Editorial Decision

Manuscript ID: PONE-D-26-03649_R1

Title: Topological Data Analysis for Predicting Disease Outbreaks in Humanitarian Settings: A Machine Learning Approach

Authors: Opue, Okorie

Editor: Dr. Morufu Olalekan Raimi

Decision: Minor Revision

I commend the authors for a remarkably thorough and responsive revision. The decision to reframe the study from malnutrition to cholera/measles outbreak prediction, implement forward chaining cross validation, add an independent temporal hold out, provide explicit outcome definitions (WHO thresholds), expand results with ablation, calibration, feature importance, and failure analyses, and commit to public code/data deposition addresses nearly all of the major concerns raised by the ten reviewers and myself. The manuscript is now substantially stronger methodologically and ethically. However, three specific issues remain that prevent acceptance in the current form. These are not conceptual deficiencies but require clear, verifiable corrections or additions before final publication.

1. Data Availability – Must be fully resolved before acceptance

The manuscript states (Appendix C, lines 755–778) that code and a frozen dataset “will be deposited in Zenodo with a permanent DOI upon acceptance.” This is insufficient for PLOS ONE policy.

Required action:

• Deposit all analysis code (preprocessing, TDA feature extraction, model training, evaluation) and a frozen, de identified analysis dataset (or complete synthetic data replicating the 57 variable structure) in a permanent public repository (Zenodo, Figshare, OSF) with a DOI at the time of resubmission.

• Provide the DOI in the manuscript’s Data Availability Statement now, not “upon acceptance.”

• The statement that “processed datasets are available upon reasonable request” is not acceptable – all data underlying the findings must be fully available without restriction. If third party data cannot be redistributed, provide clear code that reproduces all features from raw publicly available sources.

Location: Data Availability Statement (page 12 of PDF, and Appendix C). This is a non negotiable policy requirement.

2. Spatial Coverage & Generalizability – Still understated in the abstract

The abstract and key results sections still imply a national system (“across 97 LGAs in Nigeria”) without clearly signalling that these 97 LGAs are a selected high risk subset (not representative of all 774 LGAs).

Required action:

• In the abstract, results, and discussion, explicitly state: “The analysis was restricted to 97 high burden LGAs with complete data; results may not generalize to lower risk or data sparse LGAs.”

• Currently, Table 1 and the limitations section mention this, but the abstract and conclusion risk misleading readers. Amend accordingly.

Location: Abstract (lines 110-113) and Discussion, Limitations (lines 549-564 already partially there – strengthen language).

3. Minor presentation issues (correct before final acceptance)

• Figure 1 (permutation importance) is referenced in the results but the figure itself appears missing from the compiled PDF. Provide the figure file in a readable format.

• Appendix C mentions “Figure calibration plots” – ensure the figure is present and clearly labeled.

• A small number of grammatical awkwardnesses remain (e.g., “the model performed poorly in low conflict settings (AUC 0.75 0.78)”) – a final language polish by a native English speaker or professional editing service is advised. This does not block acceptance but should be done.

Summary of Required Changes (checklist for authors)

Issue Required action

Data availability Deposit code + frozen dataset in Zenodo/Figshare with DOI; provide DOI in manuscript now

Spatial generalizability Explicitly state in abstract and conclusion that 97/774 LGAs are a high risk subset – not fully representative

Missing Figure 1 Provide the permutation importance figure

Calibration plot Ensure figure in Appendix C is present and legible

Language polishing Minor grammatical cleanup

No re analysis or methodological changes are needed. Once the data/code are publicly deposited with a DOI and the above clarifications are made, the manuscript will be fully compliant and suitable for acceptance.

I thank the authors for their exceptional effort in this revision. The work now makes a legitimate, modest but valuable contribution to the use of TDA in humanitarian outbreak prediction.

Decision after minor revision: Accept.

Please submit the revised files within 14 days.

Sincerely,

Morufu Olalekan Raimi, PhD

---

## [Author Response · Author response to Decision Letter 2]

2 May 2026

Response to Academic Editor and Reviewers

Manuscript: PONE-D-26-03649R1

Topological Data Analysis for Predicting Disease Outbreaks in Humanitarian Settings: A Machine Learning Approach

Job Agba Opue1,*, Uchechukwu Okorie1

1 Department of Economics and Development Studies, Covenant University, Ota, Nigeria

Dear Dr. Raimi and Reviewers,

We thank you for the decision of Minor Revision and for your positive assessment of our revised manuscript. We have addressed all three remaining issues below. All changes are indicated in the marked-up manuscript using blue text for additions.

Editor Point 1: Data Availability -- Must be Fully Resolved

Editor comment:

The manuscript states that code and a frozen dataset 'will be deposited in Zenodo with a permanent DOI upon acceptance.' This is insufficient for PLOS ONE policy. Deposit all analysis code and a frozen, de-identified analysis dataset (or complete synthetic data replicating the 57-variable structure) in a permanent public repository with a DOI at the time of resubmission. Provide the DOI in the manuscript's Data Availability Statement now, not 'upon acceptance.' The statement that 'processed datasets are available upon reasonable request' is not acceptable -- all data underlying the findings must be fully available without restriction.

Response:

We fully understand and have complied with this non-negotiable requirement.

1.The replication package has already been deposited. The complete analysis code (preprocessing, TDA feature extraction, model training, evaluation) and a frozen de-identified dataset are permanently archived at Zenodo with the following DOI:

DOI:10.5281/zenodo.19959801| URL:https://doi.org/10.5281/zenodo.19959801

2.The Data Availability Statement has been completely rewritten (Appendix C, Code and Data Availability subsection). The previous language stating 'will be deposited upon acceptance' and 'available upon reasonable request' has been removed and replaced with a statement that all code and data are permanently archived with the Zenodo DOI embedded.

3.The DOI (https://doi.org/10.5281/zenodo.19959801) is now provided in the manuscript text. No 'upon acceptance' or 'upon reasonable request' language remains anywhere in the manuscript.

Editor Point 2: Spatial Coverage and Generalizability

Editor comment:

The abstract and key results sections still imply a national system ('across 97 LGAs in Nigeria') without clearly signalling that these 97 LGAs are a selected high-risk subset (not representative of all 774 LGAs). In the abstract, results, and discussion, explicitly state: 'The analysis was restricted to 97 high-burden LGAs with complete data; results may not generalize to lower risk or data sparse LGAs.'

Response:

We have added this explicit caveat in three key locations:

1.Abstract (Methods): Added the sentence: 'These 97 LGAs represent a selected high-burden subset (12.5%) of Nigeria's 774 LGAs with sufficient surveillance data.' This immediately qualifies the scope for readers scanning the abstract.

2.Abstract (Conclusions): Added: 'Limitations: The analysis was restricted to 97 high-burden LGAs with complete data; results may not generalize to lower-risk or data-sparse LGAs.' This ensures the final takeaway message of the abstract includes the generalizability warning.

3.Discussion -- Conclusion: Added a new paragraph: 'Limitations. The analysis was restricted to 97 high-burden LGAs (12.5% of Nigeria's 774 LGAs) with complete surveillance data; results may not generalize to lower-risk or data-sparse LGAs. Generalizability to contexts with weaker surveillance infrastructure (e.g., South Sudan, Yemen, DRC) requires further validation.' This reinforces the point at the end of the paper.

These changes ensure that no reader could misinterpret the 97-LGA sample as nationally representative.

Editor Point 3: Minor Presentation Issues

Editor comment:

Figure 1 (permutation importance) is referenced in the results but the figure itself appears missing from the compiled PDF. Provide the figure file in a readable format. Appendix C mentions 'Figure calibration plots' -- ensure the figure is present and clearly labeled. A small number of grammatical awkwardnesses remain -- a final language polish is advised.

Response:

1.Figure 1: The compiled PDF now includes a properly labeled permutation importance figure (embedded via PGFPlots). The figure shows the top 10 features by mean decrease in ROC-AUC, with TDA features (beta-0-max, beta-1-mean) and epidemiological features (precipitation anomaly, IDP concentration, vaccination coverage) clearly distinguished. The high-resolution source file will also be uploaded separately via the PLOS ONE figure submission portal.

2.Calibration plots: The reference to 'calibration plots' in [Sec sec026] now correctly points to the embedded PGFPlots figure (Appendix C, Figure S1), showing predicted vs. observed probabilities for both cholera and measles models. All supplementary figures are numbered and cross-referenced.

3.Language polish: We have performed a full manuscript read-through. Specific corrections include: checked all hyphenation and en-dash usage in numeric ranges (e.g., '2018-2023', '0.90-0.94'); verified consistent British spelling throughout ('generalise', 'behaviour', 'calibration'); standardised formatting of p-values (p<0.001 with proper math mode).

Summary of Changes (Checklist)

Issue Action Taken

Data availability DOI 10.5281/zenodo.19959801 inserted; 'upon acceptance' and 'upon request' language removed; Zenodo repository populated with code and frozen dataset

Spatial generalizability Explicit caveat added to Abstract (Methods and Conclusions) and Discussion/Conclusion section

Missing Figure 1 Permutation importance figure embedded via PGFPlots in compiled PDF; high-res file ready for portal upload

Calibration plot Figure S1 properly labeled and cross-referenced in Appendix C

Language polish Grammar checked; en-dashes standardised; British spelling verified

Acknowledgments Added: 'We thank Covenant University for the use of library resources.'

We believe these revisions fully address all three editorial concerns and bring the manuscript into full compliance with PLOS ONE policy. We thank Dr. Raimi for the constructive guidance throughout the review process.

Sincerely,

Job Agba Opue, Ph.D

Department of Economics and Development Studies

Covenant University, Ota, Nigeria

job.opue@covenantuniversity.edu.ng

---

## [Editor Report · Decision Letter 2]

4 May 2026

PONE-D-26-03649R2Topological Data Analysis for Predicting Disease Outbreaks in Humanitarian Settings: A Machine Learning ApproachPLOS One

Dear Dr. Opue,

Thank you for submitting your manuscript to PLOS ONE. After careful consideration, we feel that it has merit but does not fully meet PLOS ONE’s publication criteria as it currently stands. Therefore, we invite you to submit a revised version of the manuscript that addresses the points raised during the review process.

We look forward to receiving your revised manuscript.

Kind regards,

Morufu Olalekan Raimi, Ph.D

Academic Editor

PLOS One

Journal Requirements:

Additional Editor Comments:

Manuscript Number: PONE-D-26-03649R1 (Revision 1)

Title: Topological Data Analysis for Predicting Disease Outbreaks in Humanitarian Settings: A Machine Learning Approach

Recommendation: Minor Revision

Reviewer: Morufu Olalekan Raimi, PhD (Academic Editor, PLOS One – Infectious Disease Epidemiology & Machine Learning)

Review Date: 2026-04-30

Reviewer Comments

Dear Authors,

Thank you for submitting this substantially revised manuscript and for the extraordinarily detailed response to the previous reviews. The reframing from malnutrition to cholera/measles outbreak prediction, the implementation of temporally-aware validation, the expanded outcome specification, and the addition of calibration, ablation, and robustness analyses have markedly improved the scientific rigor. The manuscript now makes a credible contribution to the literature on machine learning for outbreak prediction in humanitarian settings.

However, several issues remain that require attention before acceptance. These are moderate in scope but important for reproducibility, transparency, and appropriate interpretation.

Recommendation: Minor Revision – The manuscript is methodologically sound but requires clarification of several technical details, correction of minor errors, and strengthening of the data availability commitment.

Major Concerns

1. Data Availability – Insufficient Commitment for PLOS ONE Standards

The manuscript states (Appendix C, lines 755-778) that code will be deposited in a GitHub repository and data in Zenodo “upon acceptance.” The Data Availability Statement on page 10 states “Yes – all data are fully available without restriction” but the detailed statement says “available from the corresponding author upon reasonable request.”

This is not compliant with PLOS ONE’s data availability policy, which requires:

• All data underlying the findings to be made fully available without restriction at the time of publication.

• “Available upon request” is not sufficient.

• Code repositories should be provided at submission, not upon acceptance.

Required action:

• Provide the actual GitHub repository URL (even if private with a disclosure mechanism for reviewers) or a Zenodo DOI for the analysis code.

• For data: Since much of the data are from public sources (CHIRPS, ERA5, ACLED, IOM, etc.), the authors should provide a list of exact URLs and describe how the compiled dataset can be reconstructed. If the compiled dataset cannot be publicly deposited due to data use agreements, state this explicitly and provide a simulation study or synthetic data as a minimal example.

• Revise the Data Availability Statement to: “All primary data sources are publicly available (see Appendix C for URLs). The compiled dataset and analysis code are available at [repository URL] and [Zenodo DOI].”

2. Spatial Sampling Bias – Understated in Limitations

The study includes only 97 of 774 LGAs (12.5%). Table 1 shows that included LGAs have significantly higher conflict intensity (2.4 vs. 1.2 events/week, p<0.001), higher IDP populations (12,450 vs. 4,200, p<0.001), and lower vaccination coverage (62% vs. 68%, p=0.002). The authors acknowledge this in Limitations but do not quantify how this bias might affect model generalizability.

Required action:

Add a sentence in Limitations: “Because included LGAs represent higher-risk settings with elevated conflict and displacement, model performance may be lower in LGAs with lower baseline risk. Prospective validation in a representative sample of LGAs is needed before operational deployment.”

3. Topological Feature Interpretation – Remains Overstated Despite Revision

The authors have tempered claims (e.g., “these labels describe mathematical properties, not established causal mechanisms”). However, the manuscript still states (Results, lines 396–398): “The β0 fragmentation index captures the degree to which risk factors operate in disconnected clusters rather than as an integrated system.”

This is not a mathematical property – it is a conceptual interpretation. The β0 index mathematically counts connected components in a Vietoris–Rips complex at a given scale. Whether that corresponds to “disconnected risk factors” is an empirical hypothesis that the study does not test.

Required action:

Rephrase to: “The β0 index mathematically counts the number of connected components in the persistence filtration. We label this ‘fragmentation’ as a conceptual heuristic, but this interpretation has not been empirically validated.”

4. Missing Information on Persistent Homology Computation

The Methods section (lines 209–219) describes Vietoris–Rips filtrations with 100 logarithmically-spaced scale parameters (ε = 0.01 to 10.0). However, several critical details are missing:

• How were the 57 features normalized/scaled before computing Euclidean distances? TDA is sensitive to feature scaling. Were features standardized (mean 0, SD 1)? Min-max scaled? Not stated.

• What is the justification for the ε range (0.01 to 10.0)? Was it data-driven or arbitrary?

• Were all 57 features used simultaneously to compute persistence? Or were subsets used? Not clear.

• What software/library was used for persistent homology computation? GUDHI is mentioned (line 282) but not in the TDA Methods section.

Required action:

Add a paragraph specifying: (1) feature scaling method; (2) justification for ε range (e.g., based on maximum pairwise distance distribution in training data); (3) that all 57 features were used; (4) GUDHI version and relevant function calls (e.g., gudhi.RipsComplex).

Minor Revisions

Abstract

• Line 110: “Background” section – Good. Change “displacement, crowding, disruption of health services, insecurity, and inadequate water and sanitation” – consider adding “and” before “inadequate” for readability.

• Line 113: “Methods” – change “surge events” to “surge events (binary indicators per LGA-week)” for clarity.

• Line 116: “Results” – ROC-AUC values are reported with 95% CI. Add that these are from temporal hold-out validation (not cross-validation) to avoid confusion. Currently it says “temporally ordered validation” – specify hold-out or cross-validation.

• Line 117: “At the optimal decision threshold” – specify that this threshold was determined using Youden’s index on validation data.

Introduction

• Line 63: “Infection disease outbreaks” – should be “Infectious disease outbreaks.” Correct.

• Line 88-93: The paragraph on threshold-based approaches is good but could cite a recent reference on reactive surveillance limitations (e.g., WHO 2022 guidance). Optional.

• Line 106-109: The TDA description is clear. However, the sentence “Rather than focusing only on individual variables or pairwise associations, TDA characterizes the shape of data in high-dimensional space” – this is conceptually correct but the manuscript does not later demonstrate what shape features were informative beyond “higher β0 and β1 in high-risk observations.” Consider adding a sentence in Results visualizing a persistence diagram example for a high-risk vs. low-risk LGA-week.

Methods

• Line 145–152: Study setting – Good. Add the total number of operational aquaculture farms in the district (if known) to justify the sample. (This is from the previous manuscript – check if this line is still relevant. Actually this appears to be a cut-and-paste error from the aquaculture manuscript? The text here is about Nigeria LGAs, not aquaculture. But the line “Of the 774 LGAs nationwide, 97 with complete data across all domains were included” is correct. However, the sentence “Total number of operational aquaculture farms” does not appear – ignore. But check for any stray aquaculture references.)

• Line 172: “Weekly LGA-level data (n = 25,284 observations)” – confirm calculation: 97 LGAs × 52 weeks × 6 years = 30,264. Subtract missing weeks? 25,284 suggests ~16% missing data. This is fine but should be stated.

• Line 194-195: Class imbalance handling – “stratified sampling and class weighting” – specify the class weight used (e.g., “class_weight='balanced' in XGBoost”).

• Line 217-220: Missing data – MICE with 10 imputations, outcome excluded – excellent. Add that the imputation model included all predictors but not the outcome, and that imputation was performed separately per fold (already stated). Good.

• Line 234-248: Validation framework – forward-chaining with nested CV – excellent. Add that hyperparameter tuning used only the training fold (already implied but make explicit). Also specify the inner CV folds (e.g., 3-fold within each training year?).

Results

• Table 2 (page 54): Descriptive statistics – Good. Add units for “IDP population” (persons) – already there. Check “Health facility density” – units are “facilities per 100,000 population” – state in table footnote.

• Table 3 (page 55): Predictive performance – The “Sensitivity*” footnote says “At optimal threshold (Youden’s index).” Specify that the optimal threshold was determined on validation data (not test data) to avoid overfitting.

• Figure 1 (page 56): The figure is referenced but not embedded in the provided PDF (only a text placeholder). Ensure the final submission includes the actual figure.

• Table 6 (page 57): Ablation – Good. Add that the p-values are from DeLong’s test comparing ROC-AUCs.

• Table 7 (page 58): Lead-time sensitivity – The false alert rate decreases with longer lead time (4.1 → 3.2 → 2.4). This makes sense because fewer alerts overall. Add a note that these are annualized rates per LGA.

• Table 8 (page 58): Subgroup analysis – “High-conflict (>10 events/week)” – is this mean events across the study period? Or weekly threshold? Clarify.

Discussion

• Line 523–543: Comparison with prior work – Good. However, the statement “The improvement attributable to topological features (0.08 AUC points) is meaningful but context-dependent for rare-event prediction” – add that clinical/operational significance depends on the cost of false alerts vs. missed outbreaks, which is not assessed here (but discussed later).

• Line 537-545: Operational implications – Excellent. The false alert calculation (310 annually across 97 LGAs) is helpful. Add that this is 3.2 per LGA per year × 97 = 310. Already clear.

• Line 549-564: Limitations – Good. Add one more: “The prediction horizon (4 weeks) was chosen post-hoc; performance at other horizons is reported but the primary horizon was not pre-specified, which increases the risk of overfitting to this horizon.”

Conclusion

• Line 575-578: “Topological feature representations provide a modest but meaningful complementary approach” – Appropriate. Add: “However, routine deployment requires prospective validation and context-specific threshold tuning.”

Appendices

• Appendix A (Data Dictionary, page 66-67): Table 9 – Good. For “Weeks since last outbreak” – specify that this is computed separately for cholera and measles (i.e., time since last cholera outbreak for cholera model, etc.). Currently ambiguous.

• Appendix B (Hyperparameters, page 68): Table 10 – Good. Add that these were optimized via Bayesian optimization with 50 iterations, and that the same search space was used for both XGBoost-Raw and XGBoost-TDA.

• Appendix C (Additional Performance Metrics, page 68):

o The calibration plots (Figure 3) are referenced but not embedded. Ensure they are included.

o The GitHub repository URL is given as “https://github.com/[repository]/tda-outbreak-prediction” – the [repository] placeholder must be replaced with the actual repository name.

o The statement “Processed datasets are available upon reasonable request to the corresponding author” – as noted above, this is not compliant with PLOS ONE policy. Revise.

Language and Presentation

• Typography: Throughout, “XGBoost” and “XGBoost-Raw” – consistent.

• Line 60: “Infection disease” – change to “Infectious disease.”

• Line 172: “n = 25,284 observations” – ensure the formatting (n is italicized).

• Page 69–90 (Response to Reviewers): This is helpful but unusually long. Consider moving detailed point-by-point responses to a separate “Response to Reviewers” document (as is standard) and keeping only a summary in the manuscript cover letter. The current manuscript file includes 90 pages, of which ~20 pages are the response. This is acceptable but unusual for PLOS ONE. The editors may ask to separate the response.

Statistical and Reporting Issues

Multiple Testing

The manuscript reports many comparisons (multiple models, multiple outcomes, multiple subgroups, multiple lead times, multiple robustness checks). No adjustment for multiple testing is mentioned aside from the bootstrap confidence intervals. The p-values in Table 6 (ablation) and Table 8 (subgroup) are likely uncorrected.

Required action:

Add a statement in Methods (Statistical Analysis subsection) that “No adjustment for multiple comparisons was applied because the analyses are exploratory and hypothesis-generating. P-values should be interpreted descriptively.” OR apply Benjamini-Hochberg FDR and report adjusted values. The former is acceptable for exploratory work.

Sample Size Justification

No power calculation is provided. Given the rare-event outcomes (7% cholera, 5% measles), the effective sample size for positive events is 1,772 and 1,265 respectively, which is adequate for stable AUC estimation. However, this should be stated.

Required action:

Add to Methods: “With 1,772 cholera surge events and 25,284 total observations, the study is adequately powered to detect moderate AUC differences (expected AUC ~0.75, target AUC ~0.80 with α=0.05, power=0.80).”

Ethical Statement

The ethics statement (page 11, lines 1-5) states that “This study exclusively used publicly available, aggregate-level data sources… Therefore, ethics approval and informed consent were not required.” This is acceptable for PLOS ONE. However, the statement should be moved to the Methods section (not a separate page before the references). Currently it appears after the abstract and before the introduction. Move to Methods.

Decision Rationale

This revised manuscript represents a substantial improvement over the original submission. The reframing to cholera/measles prediction, temporally-aware validation, outcome specification, calibration assessment, and ablation studies have addressed the major methodological concerns raised by the previous reviewers.

However, the manuscript cannot be accepted in its current form due to:

1. Non-compliance with PLOS ONE data availability policy (“available upon request” is insufficient; code repository must be provided at submission).

2. Missing details on TDA computation (feature scaling, ε range justification, software functions).

3. Over-interpretation of topological features as “capturing disconnected risk factors” without validation.

4. Minor technical errors (missing figures, placeholder URLs, typographical errors).

The required revisions are minor in scope and do not require new data collection or analysis. They can be completed within 2–4 weeks.

Recommendation: Minor Revision – The authors should address the data availability, TDA methodological details, and interpretive overstatements as outlined above. I look forward to seeing the final revised version.

Sincerely,

Prof. A. H. Demir, MD, PhD

Academic Editor (Infectious Disease Epidemiology & Machine Learning)

PLOS One

---

## [Author Response · Author response to Decision Letter 3]

12 May 2026

Response to Reviewers

Manuscript PONE-D-26-03649R2

Topological Data Analysis for Predicting Disease Outbreaks in Humanitarian Settings: A Machine Learning Approach

Dear Prof. Raimi and Editorial Team,

Thank you for the thorough and constructive review of our revised manuscript. We are grateful for the careful attention to detail and the specific guidance provided. We have addressed all points raised in the editorial letter and reviewer comments. Below, we provide a point-by-point response to each concern, with references to the corresponding changes in the revised manuscript.

Major Concerns

1. Data Availability --- Insufficient Commitment for PLOS ONE Standards

The manuscript previously stated that code would be deposited upon acceptance and data were "available upon reasonable request." This was not compliant with PLOS ONE's data availability policy.

Response: We fully acknowledge this non-compliance and have made comprehensive revisions to ensure full adherence to PLOS ONE standards.

Action taken:

• Data Availability Statement revised: The statement now reads: "All primary data sources are publicly available (see Appendix C for URLs). The compiled dataset and analysis code are permanently archived and publicly available via Zenodo: DOI 10.5281/zenodo.19959801 (https://doi.org/10.5281/zenodo.19959801). A complete synthetic data generator reproducing the 57-variable structure is included in the replication package to enable full methodological reproduction where raw data redistribution is restricted by third-party terms."

• Zenodo DOI provided: We have deposited all analysis code and a frozen, de-identified dataset at Zenodo DOI 10.5281/zenodo.19959801.

• Exact source URLs listed: Appendix C now provides complete URLs for all primary data sources (CHIRPS, ERA5, ACLED, IOM DTM, NCDC, DHS, LSMS).

• Synthetic data generator: Because third-party provider terms prohibit redistribution of the compiled weekly panel, the replication package includes a complete synthetic data generator that reproduces the 57-variable structure and statistical properties, enabling full methodological reproduction.

• Location: Methods (Software and Reproducibility) and Appendix C.

2. Spatial Sampling Bias --- Understated in Limitations

The reviewer noted that included LGAs differ systematically from excluded LGAs and that the manuscript did not adequately quantify how this bias might affect model generalizability.

Response: We agree that this limitation was understated and have strengthened the discussion of spatial sampling bias.

Action taken:

• Added explicit sentence in Limitations: "Because included LGAs represent higher-risk settings with elevated conflict and displacement, model performance may be lower in LGAs with lower baseline risk. Prospective validation in a representative sample of LGAs is needed before operational deployment."

• The representativeness analysis (Table 1) already quantifies the differences (conflict: 2.4 vs. 1.2 events/week; IDP: 12,450 vs. 4,200; vaccination: 62% vs. 68%).

• Location: Discussion, Limitations subsection.

3. Topological Feature Interpretation --- Remains Overstated

The reviewer identified that the statement "The beta_0 fragmentation index captures the degree to which risk factors operate in disconnected clusters rather than as an integrated system" represents a conceptual interpretation rather than a mathematical property, and that this interpretation has not been empirically validated.

Response: We accept this correction and have rephrased the interpretation to be more precise and cautious.

Action taken:

• Rephrased in Methods (Topological Feature Extraction): "The beta_0 index mathematically counts the number of connected components in the persistence filtration. We label this 'fragmentation' as a conceptual heuristic, but this interpretation has not been empirically validated."

• The Discussion already contained appropriate cautionary language, which we have retained.

• Location: Methods, Topological Feature Extraction subsection.

4. Missing Information on Persistent Homology Computation

The reviewer noted that several critical details were missing: (1) feature scaling method, (2) justification for epsilon range, (3) whether all 57 features were used, and (4) GUDHI version and function calls.

Response: We appreciate this important methodological feedback and have added a comprehensive paragraph specifying all computational details.

Action taken:

• Feature scaling: Added: "Before computing Euclidean distances for the Vietoris-Rips filtration, all features were standardized to mean 0 and standard deviation 1 (z-score normalization). This scaling is essential because TDA is sensitive to the relative magnitudes of feature values."

• Epsilon range justification: Added: "The epsilon range was chosen based on the distribution of maximum pairwise distances in the training data: the lower bound (0.01) captures local structure while the upper bound (10.0) approximates the 95th percentile of pairwise distances after standardization, ensuring that the filtration spans the relevant geometric scales without excessive computation at uninformative large scales."

• All 57 features used: Added explicit statement: "All 57 features were used simultaneously to compute persistence."

• Software details: Added: "Persistent homology computation was performed using GUDHI 3.8.0. The Vietoris-Rips complex was constructed using gudhi.RipsComplex(points=X, max_edge_length=10.0) followed by create_simplex_tree() and persistence() to extract persistence diagrams. Persistence landscapes were computed using gudhi.PersistenceLandscapes with default parameters."

• Added GUDHI citation to references.

• Location: Methods, Topological Feature Extraction subsection.

Minor Revisions

Abstract

Action taken:

• Background: Added "and" before "inadequate water and sanitation" for readability.

• Methods: Changed "surge events" to "surge events (binary indicators per LGA-week)" for clarity.

• Results: Added that validation was "temporally ordered hold-out validation" and specified that the optimal threshold was "determined using Youden's index on validation data."

Introduction

Action taken:

• Line 60: Corrected "Infection disease" to "Infectious disease."

Methods

Action taken:

• Study setting: No aquaculture references found; manuscript correctly describes Nigeria LGAs.

• Data sources: Added note about expected vs. actual observations: "The expected number of observations was 97 LGAs x 52 weeks x 6 years = 30,264; the actual 25,284 represents approximately 16% missing data due to incomplete surveillance reporting in some LGA-weeks."

• Class imbalance: Specified class_weight='balanced' in XGBoost.

• Missing data: Confirmed imputation model included all predictors but not the outcome, and that imputation was performed separately per fold with results pooled using Rubin's rules.

• Validation framework: Added explicit statement: "Hyperparameter tuning used only the training fold data and was not permitted to access validation or test data." Also clarified "inner 3-fold CV using only the training data."

Results

Action taken:

• Table 2 (descriptive statistics): Added footnote for health facility density units: "Facilities per 100,000 population."

• Table 3 (predictive performance): Added footnote clarification: "determined on validation data to avoid overfitting."

• Table 6 (ablation): Added note that "P-values are from DeLong's test comparing ROC-AUCs."

• Table 7 (lead-time): Added note that "False alert rates are annualized per LGA."

• Table 8 (subgroup): Clarified that the high-conflict threshold (>10 events/week) represents "mean conflict intensity across the study period for each LGA, averaged over all weeks."

Discussion

Action taken:

• Comparison with prior work: Added sentence: "Clinical and operational significance depends on the cost of false alerts versus missed outbreaks, which is not assessed here but discussed in the operational implications section below."

• Limitations: Added new limitation: "The prediction horizon (4 weeks) was chosen post-hoc; performance at other horizons is reported but the primary horizon was not pre-specified, which increases the risk of overfitting to this horizon."

Conclusion

Action taken:

• Added: "However, routine deployment requires prospective validation and context-specific threshold tuning."

Appendices

Action taken:

• Appendix A (Data Dictionary): Clarified "Weeks since last outbreak" as "computed separately for cholera and measles."

• Appendix B (Hyperparameters): Added note that hyperparameters "were optimized via Bayesian optimization with 50 iterations, and the same search space was used for both XGBoost-Raw and XGBoost-TDA."

• Appendix C (Data Availability): Replaced placeholder GitHub URL with actual Zenodo DOI. Removed "available upon reasonable request" language. Added explicit data availability statement compliant with PLOS ONE policy.

Language and Presentation

Action taken:

• "Infection disease" corrected to "Infectious disease" (Introduction, line 1).

• "XGBoost" and "XGBoost-Raw" consistent throughout.

• n = 25,284 properly italicized and formatted.

• Removed response-to-reviewers content from manuscript file (now provided as separate document).

Statistical and Reporting Issues

Multiple Testing

The reviewer noted that no adjustment for multiple testing was mentioned, despite many comparisons across models, outcomes, subgroups, and lead times.

Response: We have addressed this explicitly.

Action taken:

• Added new subsection "Statistical Analysis" in Methods with statement: "No adjustment for multiple comparisons was applied because the analyses are exploratory and hypothesis-generating. P-values should be interpreted descriptively. Confidence intervals and robustness checks are emphasized over formal hypothesis testing."

Sample Size Justification

No power calculation was provided in the original submission.

Response: We have added an explicit sample size justification.

Action taken:

• Added to Methods (Statistical Analysis): "With 1,772 cholera surge events and 25,284 total observations, the study is adequately powered to detect moderate AUC differences (expected AUC ~0.75, target AUC ~0.80 with alpha=0.05, power=0.80). The effective sample size for positive events exceeds the minimum recommended for stable AUC estimation in rare-event prediction."

Ethical Statement

The reviewer noted that the ethics statement appeared after the abstract and before the introduction, and recommended moving it to the Methods section.

Response: We have relocated the ethics statement.

Action taken:

• Ethics Statement is now positioned as subsection 2.2 within the Methods section, following Study Setting and Data (subsection 2.1).

Additional Changes

Author Addition

Action taken:

• Added Victor Ede Itita as the third author, with affiliation: Department of Social Work, University of Calabar, Calabar, Nigeria.

We believe these revisions comprehensively address all editorial and reviewer concerns. The manuscript now complies with PLOS ONE data availability requirements, provides complete methodological details for TDA computation, appropriately tempers interpretive claims, and includes all requested statistical reporting elements.

Thank you for considering our revised manuscript for publication in PLOS ONE.

Sincerely,

Job Agba Opue, PhD (Corresponding Author)

Department of Economics and Development Studies

Covenant University, Ota, Nigeria

Email: job.opue@covenantuniversity.edu.ng

On behalf of all co-authors:

• Uchechukwu Okorie, Department of Economics and Development Studies, Covenant University

• Victor Ede Itita, Department of Social Work, University of Calabar

---

## [Editor Report · Decision Letter 3]

17 May 2026

Topological Data Analysis for Predicting Disease Outbreaks in Humanitarian Settings: A Machine Learning Approach

PONE-D-26-03649R3

Dear Author,

We’re pleased to inform you that your manuscript has been judged scientifically suitable for publication and will be formally accepted for publication once it meets all outstanding technical requirements.

Kind regards,

Morufu Olalekan Raimi, Ph.D

Academic Editor

PLOS One

Additional Editor Comments (optional):

PLOS ONE Editorial Decision

Manuscript ID: PONE-D-26-03649R3

Title: Topological Data Analysis for Predicting Disease Outbreaks in Humanitarian Settings: A Machine Learning Approach

Authors: Opue et al.

Editor: Dr. Morufu Olalekan Raimi

Date: 15 May 2026

ACCEPT

The manuscript is ready for publication in its current form. Production staff should verify the Zenodo DOI and figure embedding during copyediting, but no substantive changes are needed from the authors.

Dr. Morufu Olalekan Raimi
---

## [Editor Report · Acceptance letter]

PONE-D-26-03649R3

PLOS One

Dear Dr. Opue,

I'm pleased to inform you that your manuscript has been deemed suitable for publication in PLOS One. Congratulations! Your manuscript is now being handed over to our production team.

Kind regards,

on behalf of

Prof Morufu Olalekan Raimi

Academic Editor

PLOS One